# Multiple UBX proteins reduce the ubiquitin threshold of the mammalian p97-UFD1-NPL4 unfoldase

Ryo Fujisawa, Cristian Polo Rivera, Karim PM Labib*

The MRC Protein Phosphorylation and Ubiquitylation Unit, School of Life Sciences, University of Dundee, Dundee, United Kingdom

**Abstract** The p97/Cdc48 ATPase and its ubiquitin receptors Ufd1-Npl4 are essential to unfold ubiquitylated proteins in many areas of eukaryotic cell biology. In yeast, Cdc48-Ufd1-Npl4 is controlled by a quality control mechanism, whereby substrates must be conjugated to at least five ubiquitins. Here, we show that mammalian p97-UFD1-NPL4 is governed by a complex interplay between additional p97 cofactors and the number of conjugated ubiquitins. Using reconstituted assays for the disassembly of ubiquitylated CMG (Cdc45-MCM-GINS) helicase by human p97-UFD1-NPL4, we show that the unfoldase has a high ubiquitin threshold for substrate unfolding, which can be reduced by the UBX proteins UBXN7, FAF1, or FAF2. Our data indicate that the UBX proteins function by binding to p97-UFD1-NPL4 and stabilising productive interactions between UFD1-NPL4 and K48-linked chains of at least five ubiquitins. Stimulation by UBXN7 is dependent upon known ubiquitin-binding motifs, whereas FAF1 and FAF2 use a previously uncharacterised coiled-coil domain to reduce the ubiquitin threshold of p97-UFD1-NPL4. We show that deleting the *Ubnx7* and *Faf1* genes impairs CMG disassembly during S-phase and mitosis and sensitises cells to reduced ubiquitin ligase activity. These findings indicate that multiple UBX proteins are important for the efficient unfolding of ubiquitylated proteins by p97-UFD1-NPL4 in mammalian cells.

*For correspondence:
kpmlabib@dundee.ac.uk

Competing interest: The authors declare that no competing interests exist.

## Editor's evaluation

The p97/Cdc48 AAA ATPase and its heterodimeric ubiquitin adapter Ufd1-Npl4 unfold ubiquitylated proteins, often to segregate large protein complexes such as the MCM helicase at the termination of replication. The authors now demonstrate that an important difference exists between the yeast and metazoan system. While Cdc48-Ufd1-Npl4 can target MCM7 with a relatively short ubiquitin chain (5-7 units), the metazoan p97 requires either much longer chains or one of a group of accessory proteins that have previously been connected to various p97 and Ufd1-Npl4 mediated processes. The paper presents multiple lines of experimental evidence in support of the proposed ubiquitin-chain length model.

## Introduction

The regulated unfolding of proteins is essential for many aspects of cell biology and is mediated by multiple ATPases of the AAA + protein family (*Puchades et al., 2020*; *Seraphim and Houry, 2020*; *Zhang and Mao, 2020*). These comprise hexameric rings with a central pore, in which loops from each subunit form a 'spiral staircase' around the unfolded polypeptide chain. ATP hydrolysis drives conformational changes within the ATPase hexamer, leading to translocation of the unfolded polypeptide along a 'conveyor belt' of pore loops.

p97/Cdc48/VCP is a AAA + ATPase with orthologues in all three domains of life (*Bodnar and Rapoport, 2017a*; *van den Boom and Meyer, 2018*). Eukaryotic p97/Cdc48/VCP unfolds highly stable proteins to prepare them for subsequent degradation by the proteasome (*van den Boom and Meyer, 2018*; *Verma et al., 2011*; *Wang et al., 2021*). In addition, eukaryotic p97/Cdc48/VCP plays an essential role in diverse areas of cell biology by disrupting highly stable protein complexes or extracting proteins from cellular membranes. For example, p97/Cdc48/VCP is required for disassembly of the replisome at the end of chromosome replication (*Dewar et al., 2017*; *Franz et al., 2011*; *Maric et al., 2014*; *Moreno et al., 2014*; *Sonneville et al., 2017*; *Villa et al., 2021*), release of newly synthesised protein phosphatase I from an inhibitory protein complex (*van den Boom et al., 2021*; *Weith et al., 2018*), release of DNA repair complexes from chromosomes (*Kilgas et al., 2021*; *Meerang et al., 2011*; *van den Boom et al., 2016*), and the proteasomal degradation of misfolded membrane proteins (*Ravanelli et al., 2020*; *Wolf and Stolz, 2012*; *Wu and Rapoport, 2018*). Consistent with the important roles of p97/Cdc48/VCP in eukaryotic cell biology, dominant mutations in human p97 are associated with a late-onset multisystem proteinopathy (*Johnson et al., 2010*; *Meyer and Weihl, 2014*; *Watts et al., 2004*), whilst p97 is upregulated in certain cancer types (*Li et al., 2021*; *Tsujimoto et al., 2004*) and is a target for anti-cancer therapies (*Anderson et al., 2015*; *Deshaies, 2014*; *Magnaghi et al., 2013*; *Wang et al., 2021*). p97/Cdc48/VCP associates with a wide range of adaptor proteins, many of which are still understood poorly (*Buchberger et al., 2015*; *Hänzelmann and Schindelin, 2017*). Some recognise substrates directly (*van den Boom et al., 2021*; *Weith et al., 2018*), but others contain ubiquitin-binding motifs that link p97/Cdc48/VCP to poly-ubiquitylated substrates (*Bandau et al., 2012*; *Buchberger et al., 2015*; *Stach and Freemont, 2017*). The best characterised ubiquitin adaptors of p97/Cdc48/VCP are Ufd1 and Npl4, which form a heterodimeric receptor for polyubiquitin chains that include linkages via lysine 48 (K48) of ubiquitin (*Bodnar and Rapoport, 2017a*; *Pan et al., 2021b*). Studies of yeast Cdc48-Ufd1-Npl4 have shown that the Ufd1–Npl4 complex not only helps to recruit Cdc48 to polyubiquitylated substrates but is directly involved in the first step of protein unfolding, which initiates within the ubiquitin chain rather than the substrate itself and is independent of ATP hydrolysis by Cdc48 (*Bodnar and Rapoport, 2017a*; *Ji et al., 2022*; *Twomey et al., 2019*). In this way, Ufd1-Npl4 underpin a potentially universal mechanism for the recognition and unfolding of those Cdc48 substrates that are conjugated to K48-linked ubiquitin chains, without any requirement for direct recognition of the folded substrate protein itself (*Ji et al., 2022*; *Twomey et al., 2019*). It is likely that this mechanism exploits the reduced structural stability of ubiquitin within polyubiquitin chains (*Carrion-Vazquez et al., 2003*; *Morimoto et al., 2015*).

Structural studies suggested that the yeast Ufd1–Npl4 complex uses multiple ubiquitin-binding sites (*Park et al., 2005*; *Sato et al., 2019*) to interact with up to five ubiquitins within a K48-linked chain (*Twomey et al., 2019*). The functional significance of polyubiquitin binding was further demonstrated via reconstituted assays for disassembly of the 11-subunit CMG helicase with purified proteins (*Deegan et al., 2020*), in which displacement of the ubiquitylated CMG subunit provides a readout for the number of conjugated ubiquitins that are required for substrate unfolding. Such assays showed that yeast Cdc48-Ufd1-Npl4 can only process substrates with at least five conjugated ubiquitins (*Deegan et al., 2020*). This 'five-ubiquitin threshold' represents a quality control mechanism that protects substrates from premature unfolding, by coupling Cdc48 activity to efficient ubiquitylation. Until now, the ubiquitin threshold for substrate unfolding by mammalian p97-UFD1-NPL4 had not been determined.

Although Ufd1-Npl4 are essential for yeast Cdc48 and metazoan p97/VCP to unfold ubiquitylated substrates (*Blythe et al., 2017*; *Bodnar and Rapoport, 2017b*; *Deegan et al., 2020*; *Mukherjee and Labib, 2019*), eukaryotic cells including yeast and metazoa also encode a range of other adaptor proteins, which combine ubiquitin-binding motifs with a UBX domain that associates with the amino terminus of p97/Cdc48/VCP (*Alexandru et al., 2008*; *Hänzelmann and Schindelin, 2017*; *Kloppsteck et al., 2012*; *van den Boom and Meyer, 2018*). Strikingly, studies of such ubiquitin-binding adaptors of human p97 found that UBXN7/UBXD7, FAF1, FAF2/UBXD8, and UBXN1/SAKS1 all co-purified from human cells with UFD1-NPL4 in addition to p97 (*Alexandru et al., 2008*; *Hänzelmann et al., 2011*; *Lee et al., 2013*; *Raman et al., 2015*). Moreover, in vitro studies of purified FAF1 and UBXN7 showed that both factors associate preferentially with p97-UFD1-NPL4 compared to p97 alone (*Hänzelmann et al., 2011*). These findings suggested that the mechanism or regulation of p97-UFD1-NPL4 have additional layers of complexity that remain to be explored.

Here, we show that the segregase activity of mammalian p97-UFD1-NPL4 is controlled by a combination of ubiquitin chain length and multiple UBX proteins. In contrast to yeast Cdc48-Ufd1-Npl4, reconstituted human p97-UFD1-NPL4 only unfolds substrates that are conjugated to extremely long ubiquitin chains. However, association of p97-UFD1-NPL4 with the UBX proteins UBXN7, FAF1, or FAF2 reduces the ubiquitin threshold to a comparable level to yeast Cdc48-Ufd1-Npl4, dependent upon conjugating five ubiquitins to the substrate. In this way, such UBX proteins ensure that p97-UFD1-NPL4 can unfold ubiquitylated proteins with high efficiency in mammalian cells.

## Results
### Human p97-UFD1-NPL4 has a high ubiquitin threshold

To investigate how the ability of mammalian p97-UFD1-NPL4 to unfold substrate proteins is controlled by the number of conjugated ubiquitins, we adapted a reconstituted assay for the ubiquitylation and disassembly of budding yeast CMG helicase (*Deegan et al., 2020*), using purified recombinant versions of human p97 and UFD-NPL4 (*Figure 1A, B*). Initially, we used highly efficient ubiquitylation conditions (*Deegan et al., 2020*) in which ~two to three polyubiquitin chains were conjugated to the CMG-Mcm7 subunit, linked largely but not entirely via lysine 48 (K48) of ubiquitin (*Figure 1—figure supplement 1A*). The ubiquitylated helicase was then bound to beads that were coated with antibodies to the Cdc45 subunit, before incubation with human p97 and UFD1-NPL4 in the presence of ATP. Unfolding of ubiquitylated Mcm7 led to its release into the supernatant, together with all the other CMG subunits except for Cdc45 that remained bound to the antibody-coated beads (*Figure 1C*, compare lanes 1–2 and 3–4). Helicase disassembly required UFD1-NPL4 (*Figure 1C*, lanes 5 and 6) and was dependent upon ATP binding by the D1 and D2 ATPase domains of p97 (*Figure 1D, E*, Walker A mutations) together with ATP hydrolysis by the D2 ATPase (*Figure 1D, E*, Walker B mutation), as predicted by previous studies of the Cdc48/p97 unfoldase enzyme (*Blythe et al., 2017*; *Bodnar and Rapoport, 2017b*).

Subsequently, we compared the ability of human p97-UFD1-NPL4 and yeast Cdc48-Ufd1-Npl4 to disassemble CMG complexes that had a single K48-linked chain of up to ~12 ubiquitins conjugated to CMG-Mcm7 (*Figure 1—figure supplement 1B* and *Figure 2A, B*, 2.5 nM E2 + 1 nM E3). As described previously (*Deegan et al., 2020*), yeast Cdc48-Ufd1-Npl4 disassembled CMG complexes with five or more ubiquitins conjugated to CMG-Mcm7, though disassembly was most efficient when the attached ubiquitin chains were much longer (*Figure 2B*, compare lanes 9–10 and 11–12). In contrast, although p97-UFD1-NPL4 rapidly disassembled CMG complexes with very highly ubiquitylated CMG-Mcm7 (*Figure 2B*, lanes 17–18 and *Figure 2—figure supplement 1*, lanes 7–9), the human enzyme was almost inactive against CMG complexes with up to ~12 ubiquitins conjugated to CMG-Mcm7 (*Figure 2B*, lanes 15–16 and *Figure 2—figure supplement 1*, lanes 1–3; also see below, Figure 4C, lanes 3–4), even in the presence of a large excess of unfoldase (*Figure 2—figure supplement 2*). These findings indicated that the yeast and human unfoldases have important differences in their requirements for processing ubiquitylated substrates.

We then tested whether the preference of human p97-UFD1-NPL4 for highly ubiquitylated CMG was dependent on the attachment of multiple ubiquitin chains to CMG-Mcm7, since branched K48-linked ubiquitin chains were previously found to enhance the activity of human p97-UFD1-NPL4 (*Blythe et al., 2017*). However, CMG disassembly was equally efficient under conditions where a single very long ubiquitin chain was attached to CMG-Mcm7 (*Figure 1—figure supplement 1C*; *Figure 2—figure supplement 3A*). We also considered that the ability of p97-UFD1-NPL4 to disassemble highly ubiquitylated CMG was dependent upon the formation of mixed ubiquitin chains, which have been found to associate with p97 complexes in human cells (*Yau et al., 2017*). However, both human p97-UFD1-NPL4 and yeast Cdc48-Ufd1-Npl4 were able to disassemble highly ubiquitylated CMG complexes when the ubiquitin chains were exclusively K48 linked (*Figure 2—figure supplement 3B*, compare release of Mcm6 and Sld5 into the supernatant, in reactions with 'K48-only' ubiquitin). Therefore, these data indicate that human p97-UFD1-NPL4 can disassemble CMG complexes with very long K48-linked ubiquitin chains attached to CMG-Mcm7, regardless of the presence of multiple ubiquitin chains or other forms of ubiquitin chain linkage. In contrast, human p97-UFD1-NPL4 fails to disassemble CMG complexes with a single K48-linked chain of up to about 15 ubiquitins on CMG-Mcm7, contrasting with the five-ubiquitin threshold of yeast Cdc48-Ufd1-Npl4.

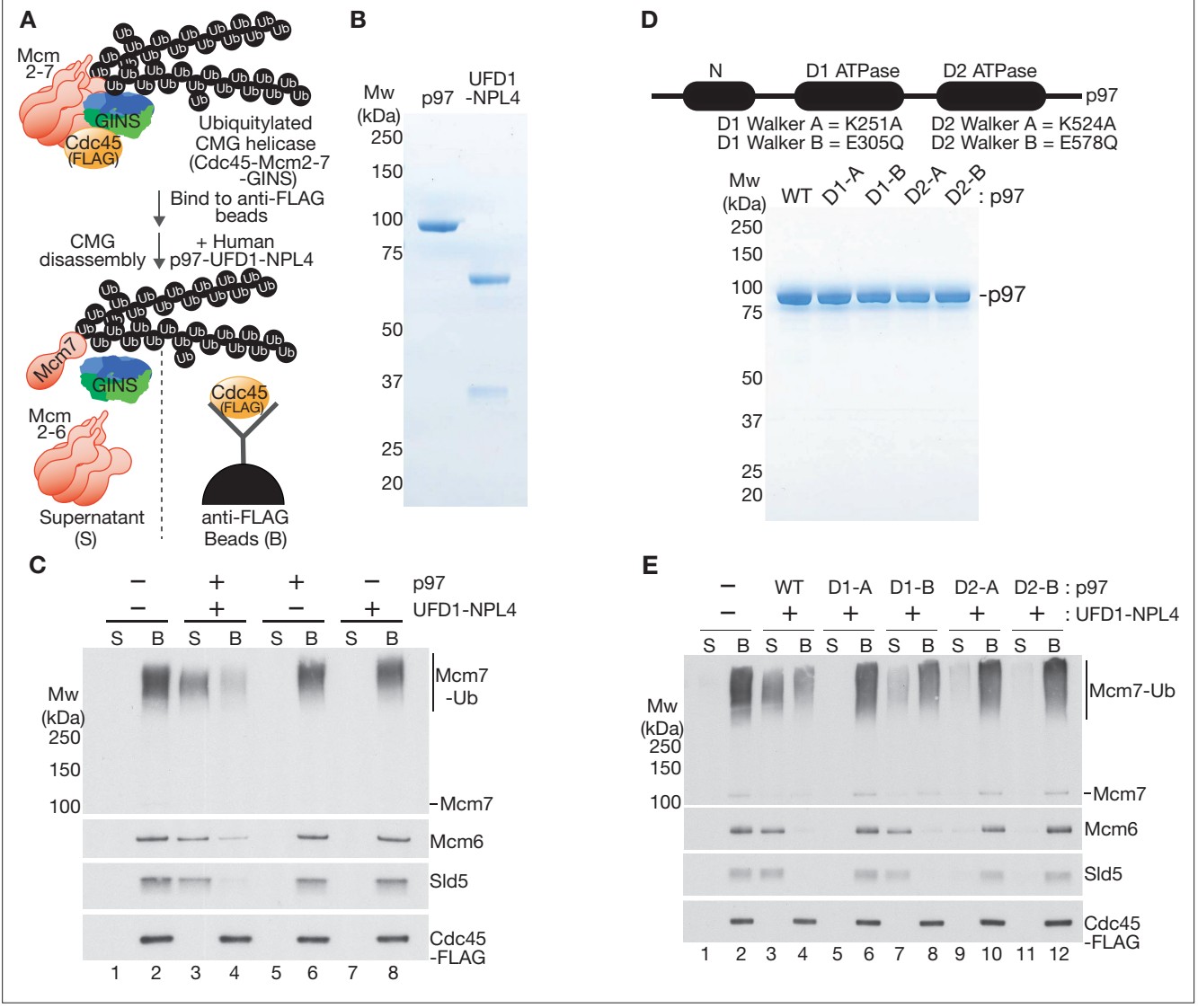

**Figure 1.** Reconstituted disassembly of ubiquitylated CMG helicase by purified human p97-UFD1-NPL4. (**A**) Budding yeast CMG helicase (with an internal FLAG-tag on the Cdc45 subunit) was ubiquitylated in vitro, bound to beads coated with anti-FLAG antibody and then incubated with recombinant human p97-UFD1-NPL4 (see Materials and methods). Disassembly of ubiquitylated CMG displaced all subunits except Cdc45 into the supernatant. (**B**) Purified human p97 and UFD1-NPL4. (**C**) CMG disassembly reactions carried out according to the scheme in (**A**), in the presence of the indicated factors. (**D**) Purified recombinant human p97 with the indicated mutations in the D1 and D2 ATPase modules. (**E**) CMG disassembly reactions equivalent to those in (**C**), in the presence of the indicated factors.

The online version of this article includes the following source data and figure supplement(s) for figure 1:

**Source data 1.** Source data for *Figure 1*.

**Figure supplement 1.** Characterisation of ubiquitin chain formation under the conditions used in this study.

**Figure supplement 1—source data 1.** Source data for *Figure 1—figure supplement 1*.

## The UFD1-NPL4 adaptor complex sets the high ubiquitin threshold of human p97

To explore whether the high ubiquitin threshold of human p97-UFD1-NPL4 reflects the properties of p97 or its UFD1-NPL4 adaptor complex, we took advantage of the high sequence conservation between human p97 and yeast Cdc48 (*Figure 3—figure supplement 1A*) and performed reconstituted CMG disassembly reactions in which the unfoldase enzyme from one species was mixed with the UFD1-NPL4 adaptor from the other. Strikingly, human UFD1-NPL4 supported the disassembly of

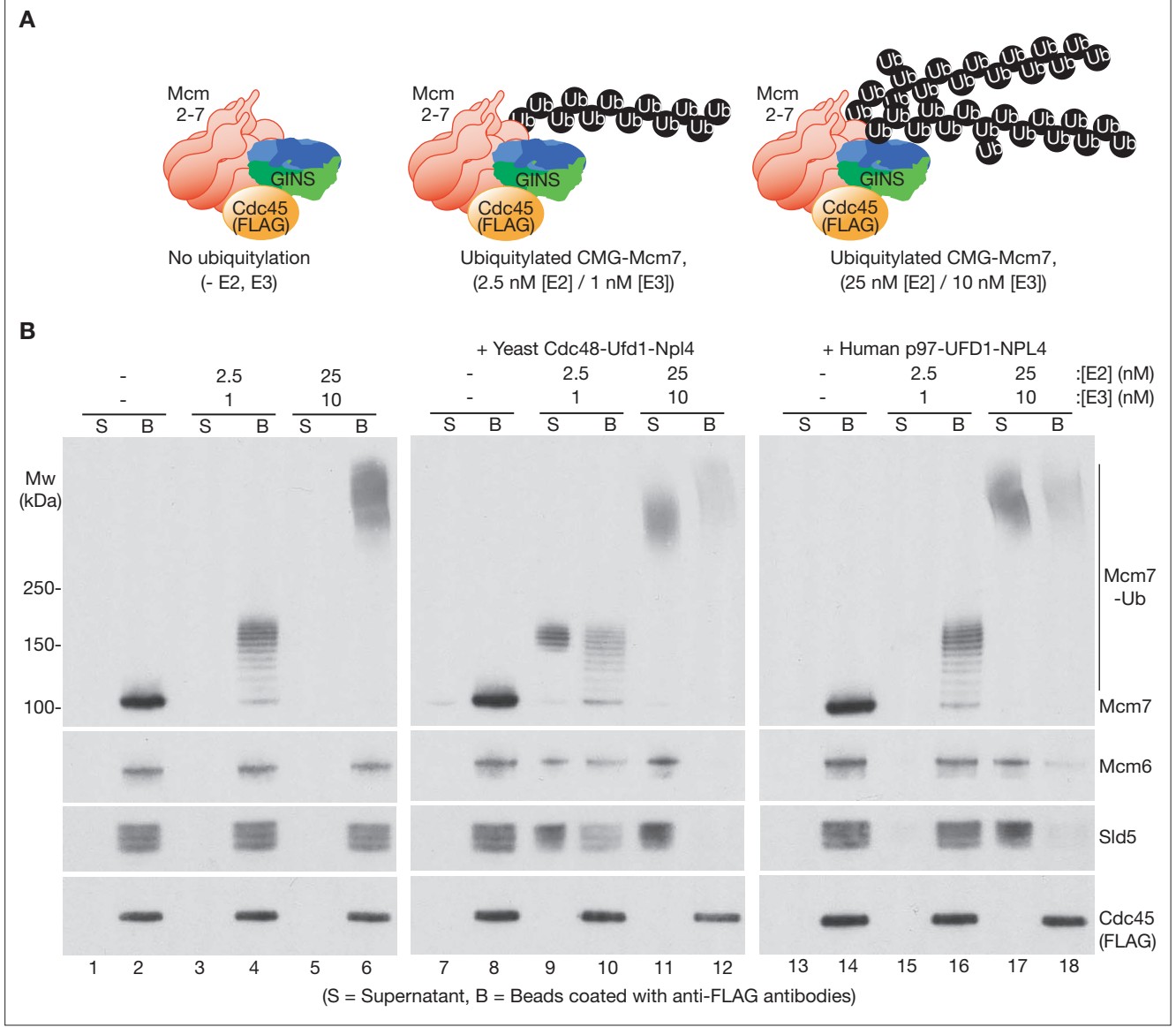

**Figure 2.** Human p97-UFD1-NPL4 has a much higher ubiquitin threshold than yeast Cdc48-Ufd1-Npl4. (**A**) Reconstituted CMG ubiquitylation reactions were performed under conditions that produced a single chain of up to ~12 ubiquitins on CMG-Mcm7 (2.5 nM E2, 1 nM E3; see *Figure 1—figure supplement 1B*), or 2–3 chains per CMG-Mcm7 (25 nM E2, 10 nM E3; see *Figure 1—figure supplement 1A*). (**B**) The products of the ubiquitylation reactions in (**A**) were bound to beads coated with anti-FLAG antibodies, before incubating as indicated with yeast Cdc48-Ufd1-Npl4 or human p97-UFD1-NPL4. CMG disassembly was monitored via displacement of subunits from beads (B) to supernatant (S), except for FLAG-tagged Cdc45 that remained bound to the anti-FLAG beads.

The online version of this article includes the following source data and figure supplement(s) for figure 2:

**Source data 1.** Source data for *Figure 2*.

**Figure supplement 1.** Time course analysis of CMG disassembly reactions by human p97-UFD1-NPL4.

**Figure supplement 1—source data 1.** Source data for *Figure 2—figure supplement 1*.

**Figure supplement 2.** Human p97-UFD1-NPL4 preferentially disassembles extensively ubiquitylated CMG helicase.

**Figure supplement 2—source data 1.** Source data for *Figure 2—figure supplement 2*.

**Figure supplement 3.** Disassembly of CMG helicase by human p97-UFD1-NPL4 does not require multiple ubiquitin chains, or other ubiquitin chain linkages apart from K48.

**Figure supplement 3—source data 1.** Source data for *Figure 2—figure supplement 3*.

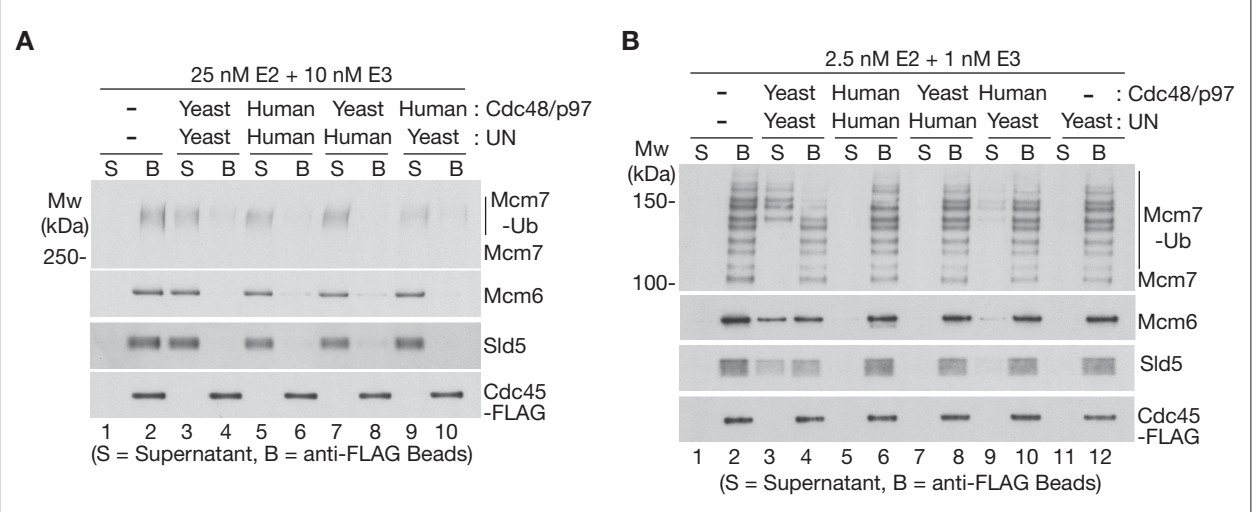

**Figure 3.** The Ufd1-Npl4/UFD1-NPL4 adaptor complex determines the ubiquitin threshold of Cdc48/p97. (**A**) CMG helicase was ubiquitylated as in *Figure 1—figure supplement 1A*, then bound to anti-FLAG beads. Disassembly reactions were then performed as indicated in the presence of yeast Cdc48 or human p97, combined with yeast Ufd1-Npl4 or human UFD1-NPL4. (**B**) Analogous reactions with CMG ubiquitylated as in *Figure 1—figure supplement 1B*.

The online version of this article includes the following source data and figure supplement(s) for figure 3:

**Source data 1.** Source data for *Figure 3*.

**Figure supplement 1.** Sequence alignment of human and yeast orthologues of p97 and UFD1-NPL4.

**Figure supplement 2.** The NPL4 groove is essential for disassembly of ubiquitylated CMG by p97-UFD1-NPL4, whereas the NZF domain is largely dispensable.

**Figure supplement 2—source data 1.** Source data for *Figure 3—figure supplement 2*.

heavily ubiquitylated CMG, not only by human p97 (*Figure 3A*, lanes 5–6), but also by yeast Cdc48 (*Figure 3A*, lanes 7–8). Similarly, yeast Ufd1-Npl4 promoted the disassembly of heavily ubiquitylated CMG by both yeast Cdc48 (*Figure 3A*, lanes 3–4) and human p97 (*Figure 3A*, lanes 9–10). These findings reflected the conserved interactions between human NPL4/yeast Npl4 and the amino terminus of human p97/yeast Cdc48 (*Bodnar et al., 2018*; *Pan et al., 2021a*; *Pan et al., 2021b*).

We then performed similar reactions with a CMG substrate that had a single ubiquitin chain of up to ~10 ubiquitins conjugated to the Mcm7 subunit (*Figure 3B*). Importantly, yeast Ufd1-Npl4 supported the disassembly of CMG complexes with five or more ubiquitins conjugated to CMG-Mcm7, not only by yeast Cdc48 (*Figures 3B and 4*), but also in the presence of human p97 (*Figure 3B*, lanes 9–10; yeast Ufd1-Npl4 was less efficient in combination with human p97 compared to reactions with yeast Cdc48, yet in both cases was selective for CMG-Mcm7 with five or more conjugated ubiquitins). In contrast, neither yeast Cdc48 nor human p97 disassembled CMG with 5–10 ubiquitins in reactions containing human UFD1-NPL4 (*Figure 3B*, lanes 7–8 and 5–6). These findings indicated that human UFD1-NPL4 and yeast Ufd1-Npl4 determine the ubiquitin threshold of their cognate unfoldase enzymes.

It is likely that human UFD1-NPL4 and yeast Ufd1-Npl4 employ a similar mechanism to initiate the unfolding of ubiquitylated substrates by p97/Cdc48. Both adaptor complexes bind to the amino terminal face of human p97/yeast Cdc48, with Npl4 forming a tower adjacent to the central p97 pore within which unfolded peptides are translocated (*Bodnar et al., 2018*; *Pan et al., 2021a*; *Pan et al., 2021b*). Moreover, human UFD1-NPL4 and yeast Ufd1-Npl4 utilise conserved ubiquitin-binding modules to recognise K48-linked ubiquitin chains, via multiple contacts including the top of the NPL4 tower and the UT3 domain of UFD1 (*Ji et al., 2022*; *Pan et al., 2021b*; *Park et al., 2005*; *Pye et al., 2007*; *Sato et al., 2019*; *Twomey et al., 2019*; *Ye et al., 2003*). Studies of the yeast unfoldase showed that binding of K48-linked ubiquitin chains to Ufd1-Npl4 led to unfolding of a ubiquitin moiety, which then interacts with a groove on the surface of the Npl4 tower before entering the central pore of Cdc48 (*Twomey et al., 2019*). Correspondingly, mutations of conserved residues in the Npl4 groove

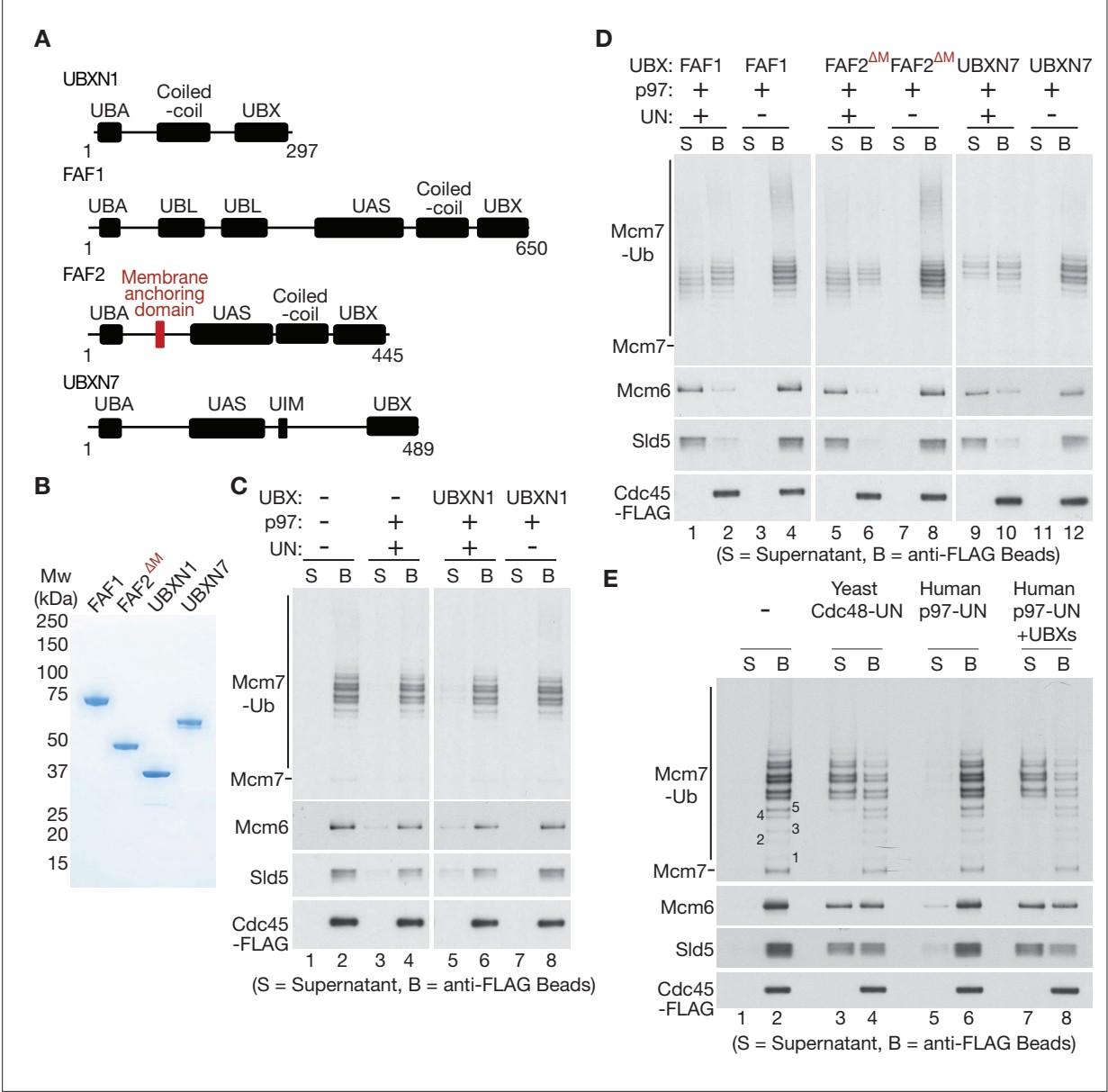

**Figure 4.** UBXN7, FAF1, and FAF2 reduce the ubiquitin threshold of p97-UFD1-NPL4. (**A**) Domain organisation of the indicated UBX proteins. UBA = 'UBiquitin-Associated' domain that binds ubiquitin; UBX = 'UBiquitin regulatory X' domain that binds p97; UBL = 'UBiquitin-Like' domain that binds ubiquitin and NEDD8; UAS = domain of unknown function in FAF1/FAF2/UBXN7. (**B**) Purified proteins – the membrane anchoring domain of FAF2 was deleted in FAF2^ΔM to facilitate expression of a soluble protein. (**C–E**) CMG disassembly reactions in the presence of the indicated factors, performed as described in *Figures 1–3*. See also *Figure 4—figure supplements 1–3*.

The online version of this article includes the following source data and figure supplement(s) for figure 4:

**Source data 1.** Source data for *Figure 4*.

**Figure supplement 1.** Purified human UBX proteins bind to p97-UFD1-NPL4.

**Figure supplement 1—source data 1.** Source data for *Figure 4—figure supplement 1*.

**Figure supplement 2.** Stimulation of p97-UFD1-NPL4 activity by UBX proteins requires the NPL4 groove and suppresses loss of the ubiquitin-binding NPL4-NZF.

**Figure supplement 2—source data 1.** Source data for *Figure 4—figure supplement 2*.

**Figure supplement 3.** Human UBX proteins stimulate CMG helicase disassembly by yeast Cdc48 in the presence of human UFD1-NPL4.

**Figure supplement 3—source data 1.** Source data for *Figure 4—figure supplement 3*.

blocked substrate unfolding by yeast Cdc48-Ufd1-Npl4 (*Twomey et al., 2019*). We found that mutation of the equivalent residues in human NPL4 (*Figure 3—figure supplement 2A, B*, NPL4-AAE) blocked the disassembly of ubiquitylated CMG helicase (*Figure 3—figure supplement 2C*, compare lanes 3–4 and 7–8), without affecting the association of NPL4 with UFD1 (*Figure 3—figure supplement 2B*), or the interaction of UFD1-NPL4 with p97 (*Figure 3—figure supplement 2D*) or folded K48-linked polyubiquitin (*Figure 3—figure supplement 2E*). These data suggested that human p97-UFD1-NPL4 also initiates substrate unfolding via an unfolded ubiquitin intermediate, analogous to the action of yeast Cdc48-Ufd1-Npl4.

However, yeast Ufd1-Npl4 and human UFD1-NPL4 are considerably more divergent in primary sequence, compared to yeast Cdc48 and human p97 (*Figure 3—figure supplement 1A, C*), suggesting some degree of functional diversification. Moreover, human NPL4 has a ubiquitin-binding Zinc finger ('NPL4-like Zinc Finger' or NZF) at the carboxy-terminal end of the protein, which in vitro represents the highest affinity binding site for ubiquitin in human NPL4 (*Meyer et al., 2002*; *Tsuchiya et al., 2017*) and is not present in the yeast orthologue. Consistent with previous studies (*Meyer et al., 2002*), deletion of the NPL4-NZF did not impair association with UFD1 of p97 (*Figure 3—figure supplement 2B, D*) but largely abrogated the interaction of UFD1-NPL4 with K48-linked ubiquitin chains in vitro (*Figure 3—figure supplement 2D*). Nevertheless, the NPL-NZF was previously found to be dispensable in human cells for the degradation of misfolded proteins in the endoplasmic reticulum (*Ye et al., 2003*), a process known as ERAD that is dependent upon ubiquitylation and the p97-UFD1-NPL4 unfoldase. This suggested that the evolutionarily conserved low affinity binding sites for K48-linked ubiquitin chains in human UFD1-NPL4, at the top of the NPL4 tower and in the UT3 domain of UFD1 (*Ji et al., 2022*; *Pan et al., 2021b*; *Park et al., 2005*; *Pye et al., 2007*; *Sato et al., 2019*; *Twomey et al., 2019*; *Ye et al., 2003*), are sufficient to initiate the unfolding of ubiquitylated substrates of p97. Indeed, we found that human p97-UFD1-NPL4-ΔNZF still supported the disassembly of heavily ubiquitylated CMG helicase in reconstituted in vitro assays, albeit with reduced efficiency compared to wild-type NPL4 (*Figure 3—figure supplement 2C*, compare lanes 1–6). Considered together with the data and previous observations discussed above, these findings indicated that the dynamic association of the NPL4 tower and UFD1-UT3 domain of human UFD1-NPL4 with very long K48-linked ubiquitin chains is sufficient to initiate substrate unfolding, dependent upon the NPL4 groove. Additional ubiquitin binding by the NPL4-NZF domain is not an essential part of the mechanism of substrate unfolding, but likely increases the efficiency of substrate engagement by human p97-UFD1-NPL4, as does increased length of the K48-linked ubiquitin chains.

## UBXN7, FAF1, and FAF2 all reduce the ubiquitin threshold of mammalian p97-UFD1-NPL4

To process substrate proteins that are conjugated to shorter K48-linked ubiquitin chains, it is possible that p97-UFD1-NPL4 is aided in mammalian cells by additional adaptor proteins. In this regard, it is interesting that the UBX proteins FAF1, FAF2, UBXN7, and UBXN1 were previously found to co-purify with ubiquitin chains, p97 and UFD1-NPL4 from human cell extracts (*Alexandru et al., 2008*). Moreover, we found that all four factors bound directly to p97-UFD1-NPL4 in vitro (*Figure 4—figure supplement 1*; the association of each UBX protein with UFD1-NPL4 was dependent upon the presence of p97). Nevertheless, the interaction of UBXN1 with p97-UFD1-NPL4 was relatively weak, both in vitro with recombinant proteins (*Figure 4—figure supplement 1*) and in human cells (*Alexandru et al., 2008*).

To monitor the impact of UBXN1, FAF1, FAF2, and UBXN7 on the unfoldase activity of human p97-UFD1-NPL4, we purified the four UBX proteins (*Figure 4A, B*) and added them to reconstituted CMG disassembly assays. Reactions were performed under conditions where most CMG complexes had more than five ubiquitins conjugated to the Mcm7 subunit (*Figure 4C, D*), principally as K48-linked ubiquitin chains (*Figure 1—figure supplement 1D*) that were too short to support CMG disassembly by p97-UFD1-NPL4 alone (*Figure 4C*, lanes 1–4). UBXN1 did not significantly alter the in vitro activity of the p97-UFD1-NPL4 unfoldase (*Figure 4C*), consistent with the weak interaction of UBXN1 with p97-UFD1-NPL4. In contrast, FAF1, FAF2, and UBXN7 all stimulated the disassembly of ubiquitylated CMG by human p97 (*Figure 4D*, lanes 1–2, 5–6, and 9–10). In each case, disassembly was still dependent upon the presence of UFD1-NPL4 (*Figure 4D*, lanes 3–4, 7–8, and 11–12). These data indicated

that FAF1, FAF2, and UBXN7 stimulate the unfoldase activity of p97-UFD1-NPL4 when the K48-linked ubiquitin chains attached to substrate are of limiting length.

Strikingly, helicase disassembly by human p97-UFD1-NPL4 in the presence of the three UBX proteins required five or more ubiquitins on the CMG-Mcm7 subunit (*Figure 4E*), thereby matching the minimal ubiquitin threshold of yeast Cdc48-Ufd1-Npl4 (*Deegan et al., 2020*) that is thought to reflect the presence of multiple conserved ubiquitin-binding domains within yeast Ufd1-Npl4 (*Park et al., 2005*; *Sato et al., 2019*; *Twomey et al., 2019*). This suggested that the UBX proteins function by stimulating productive interactions between the multiple ubiquitin-binding modules of human p97-UFD1-NPL4 and ubiquitylated substrate proteins. The initial engagement of UFD1-NPL4 with ubiquitylated substrates in the presence of UBX proteins should then lead to the trapping of an unfolded ubiquitin intermediate on the NPL4 groove. Consistent with this view, CMG disassembly in the presence of UBXN7, FAF1, and FAF2 was still dependent upon conserved residues in the NPL4 groove (*Figure 4—figure supplement 2B*; compare lanes 5–6 and 13–14).

The UBX proteins suppressed the partial defect in disassembling heavily ubiquitylated CMG that was observed in the absence of the ubiquitin-binding Zinc finger at the carboxyl terminus of NPL4 (*Figure 4—figure supplement 2*, compare release of Mcm6 and Sld5 into the supernatant in lanes 5–10). We also found that the human UBX proteins restored the ability of yeast Cdc48 to disassemble CMG complexes in the presence of human UFD1-NPL4, when the K48-linked ubiquitin chains on CMG-Mcm7 were otherwise too short to allow disassembly (*Figure 4—figure supplement 3*, lanes 1–6). Overall, these findings indicated that UBXN7, FAF1, and FAF2 augment the ability of human UFD1-NPL4 to initiate the unfolding of ubiquitylated substrates by p97.

## The UBX, UBA, and UIM domains of UBXN7 all contribute to stimulation of p97-UFD1-NPL4 activity

To explore further how UBXN7 stimulates the unfolding of ubiquitylated CMG-Mcm7 by p97-UFD1-NPL4, we purified a series of truncated versions (*Figure 5A, B*). As predicted, all the truncated proteins still associated with p97-UFD1-NPL4, except for the version lacking the UBX domain (*Figure 5—figure supplement 1A, B*) that binds to the amino terminus of p97 (*Li et al., 2017*). We then tested the ability of the truncated UBXN7 proteins to support CMG disassembly by p97-UFD1-NPL4 (*Figure 5C*). The UBX domain of UBXN7 was essential for p97-UFD1-NPL4 to disassemble ubiquitylated CMG (*Figure 5C*, compare lanes 1–4), indicating that UBXN7 functions in direct association with p97-UFD1-NPL4. Moreover, efficient CMG disassembly was dependent upon both the ubiquitin-binding UBA domain (*Figure 5C*, lanes 5–6) and also the UIM domain (*Figure 5C*, lanes 7–8) that binds to both ubiquitin and NEDD8 (*den Besten et al., 2012*; *Fisher et al., 2003*; *Young et al., 1998*). Since the reconstituted CMG disassembly reactions lack NEDD8, these findings indicated that UBXN7 uses its UBA and UIM domains to stabilise productive interactions between the p97-UFD1-NPL4 unfoldase and its ubiquitylated substrate, thereby promoting the initiation of substrate unfolding via the NPL4 groove.

## The UBX and coiled-coil domains of FAF1 and FAF2 are required to stimulate p97-UFD1-NPL4 unfoldase activity

We then purified an equivalent series of truncated versions of FAF1 (*Figure 5D, E*), all of which still associated with p97-UFD1-NPL4 except for FAF1-ΔUBX (*Figure 5—figure supplement 1C, F*). As above, the UBX domain was essential to stimulate segregase activity (*Figure 5F*, compare lanes 1–4), indicating that FAF1 functions in a complex with p97-UFD1-NPL4, similar to UBXN7. Surprisingly, however, the UBA domain of FAF1 was dispensable for stimulation of p97-UFD1-NPL4 (*Figure 5F*, FAF1-Δ268, lanes 5–6). Moreover, a carboxy-terminal fragment of FAF1 that comprised the coiled-coil and UBX domains was just as effective as the full-length protein in promoting disassembly of ubiquitylated CMG by human p97-UFD1-NPL4 (*Figure 5F*, FAF1-Δ480, lanes 7–8). In contrast, the UBX domain on its own was inactive (*Figure 5F*, FAF1-UBX, lanes 9–10). We note that the coiled coil is not required for the UBX domain of FAF1 to interact with p97-UFD1-NPL4 (*Figure 5—figure supplement 1E, F*). Furthermore, human p97 associates with UFD1-NPL4 in the absence of FAF1 or other UBX proteins (*Figure 3—figure supplement 2D*), as shown previously (*Meyer et al., 2002*), and FAF1 does not associate detectably with UFD1-NPL4 in the absence of p97 (*Figure 5—figure supplement 1F*, lane

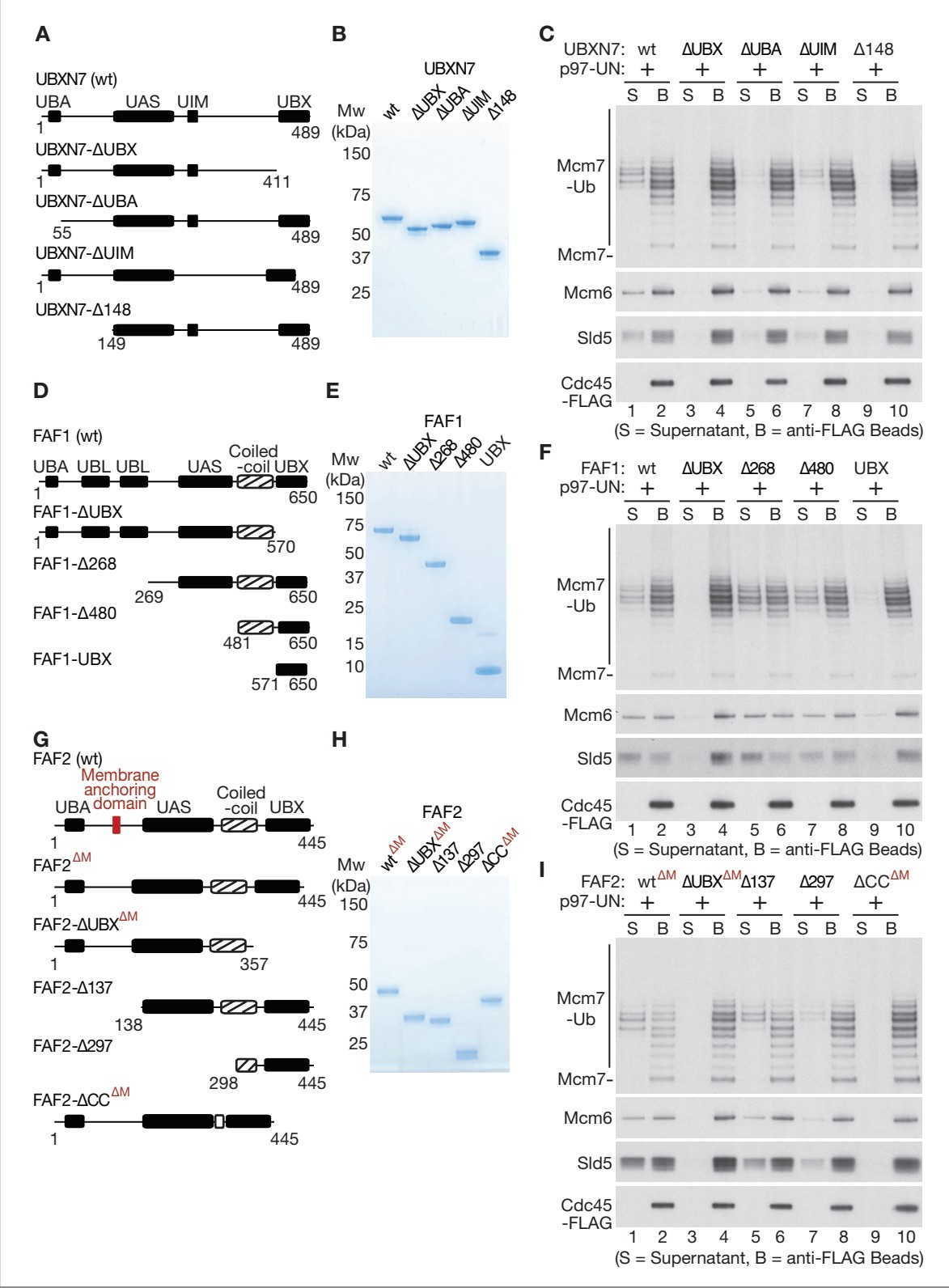

**Figure 5.** Mapping domains of UBXN7, FAF1, and FAF2 that stimulate the unfoldase activity of p97-UFD1-NPL4. (**A**) Truncations of UBXN7. (**B**) Purified proteins. (**C**) CMG was ubiquitylated as in *Figure 1—figure supplement 1D* and then bound to anti-FLAG beads. Disassembly reactions were performed as above, in the presence of the indicated factors. (**D–F**) Equivalent analysis for FAF1. (**G–I**) Analogous truncations of FAF2 – 'ΔM' indicates alleles that contain the amino terminus of the protein but lack the membrane anchoring domain. For (**A**), (**D**), and (**G**), the numbers correspond to

*Figure 5 continued on next page*

*Figure 5 continued*

residues in the full-length proteins. Domains were predicted using the SMART algorithm (http://smart.embl-heidelberg.de/) and Alphafold (*Jumper et al., 2021*).

The online version of this article includes the following source data and figure supplement(s) for figure 5:

**Source data 1.** Source data for *Figure 5*.

**Figure supplement 1.** Association of UBXN7 and FAF1 truncations with p97-UFD1-NPL4.

**Figure supplement 1—source data 1.** Source data for *Figure 5—figure supplement 1*.

**Figure supplement 2.** The coiled-coil and UBX domains of *C.elegans* UBXN-3 are required to stimulate the unfoldase activity of CDC-48_UFD-1_NPL-4.

**Figure supplement 2—source data 1.** Source data for *Figure 5—figure supplement 2*.

**Figure supplement 3.** Ubiquitin binding of human FAF1.

**Figure supplement 3—source data 1.** Source data for *Figure 5—figure supplement 3*.

1). These findings indicated that the coiled-coil domain of FAF1 has a previously unanticipated role in stimulating the unfoldase activity of p97-UFD1-NPL4.

We also generated analogous truncations of FAF2 (*Figure 5G, H*) and again found that the UBX domain was essential for stimulation of the p97-UFD1-NPL4 unfoldase (*Figure 5I*, compare lanes 1–4), whereas the UBA domain was dispensable (*Figure 5I*, FAF2-Δ137, lanes 5–6). Furthermore, a FAF2 allele that lacked the coiled coil was inactive (*Figure 5I*, FAF2-ΔCC$^{\Delta M}$, lanes 9–10), whilst a fragment comprising the coiled-coil and UBX domains retained activity (*Figure 5I*, FAF2-Δ297, lanes 7–8). These findings indicated that FAF2 and FAF1 share a common mechanism by which they stimulate the disassembly of ubiquitylated CMG by p97-UFD1-NPL4.

Finally, we tested whether the novel role for the coiled-coil domain of mammalian FAF1 and FAF2 is also conserved in the single *C. elegans* orthologue UBXN-3. In reconstituted reactions analogous to those described above, we found that a carboxy-terminal fragment of UBXN-3 comprising the coiled-coil and UBX domains was sufficient to promote the disassembly of ubiquitylated CMG helicase by *C. elegans* CDC-48_UFD-1_NPL-4 (*Figure 5—figure supplement 2A, B*, UBXN-3-Δ435). Moreover, the UBX domain was essential for UBXN-3 to stimulate CDC-48_UFD-1_NPL-4 (*Figure 5—figure supplement 2A, B*, UBXN-3-ΔUBX), as reported recently (*Xia et al., 2021*), but was insufficient in the absence of the coiled-coil domain (*Figure 5—figure supplement 2A, B*, UBXN-3-Δ527). Therefore, the coiled-coil domain is essential for UBXN-3 to stimulate *C. elegans* CDC-48_UFD-1_NPL-4, mirroring the importance of the coiled-coil domains of mammalian FAF1 and FAF2. These findings indicate that metazoan FAF1/FAF2/UBXN-3 stimulate p97-UFD1-NPL4 activity by a common mechanism, involving direct binding to the unfoldase and an essential role for the coiled-coil domain of the UBX cofactor.

## Cells lacking FAF1 and UBXN7 show defects in CMG helicase disassembly and are sensitive to inhibition of cullin ligase activity

Our data show that human p97-UFD1-NPL4 has a very high ubiquitin threshold, below which the unfoldase is dependent upon one of multiple UBX proteins, including UBXN7, FAF1, and FAF2. Such factors might act redundantly in mammalian cells, or else might stimulate p97-UFD1-NPL4 activity in distinct subcellular localisations. Previous work indicated that UBXN7 is largely nuclear in human cells (*Raman et al., 2015*). In contrast, FAF1 is present in both nucleus and cytoplasm (*Raman et al., 2015*), whereas FAF2 is a membrane anchored protein that localises to the endoplasmic reticulum, lipid droplets and the nuclear outer membrane (*Go et al., 2021*; *Raman et al., 2015*; *Zehmer et al., 2009*).

The function of human p97 and its cofactors is likely to be very highly conserved in other mammalian species, and we note that the mouse orthologues of p97-UFD1-NPL4, UBXN7, FAF1, and FAF2 are 97–100% identical to their human equivalents (*Figure 6—figure supplements 1–2*). To begin to assay for functional redundancy amongst the set of UBX proteins that stimulate mammalian p97-UFD1-NPL4 activity, we took advantage of the fact that mouse embryonic stem cells (mouse ES cells) were recently established (*Villa et al., 2021*) as a model system for studying mammalian CMG helicase disassembly. As in other metazoan species, mouse CMG is ubiquitylated on its MCM7 subunit during DNA replication termination, by the cullin ubiquitin ligase CUL2$^{LRR1}$ (*Villa et al., 2021*). Ubiquitylated CMG is then disassembled very rapidly by p97-UFD1-NPL4, but the ubiquitylated helicase can be stabilised by treating cells with a small molecule inhibitor of the p97 ATPase (*Villa et al., 2021*).

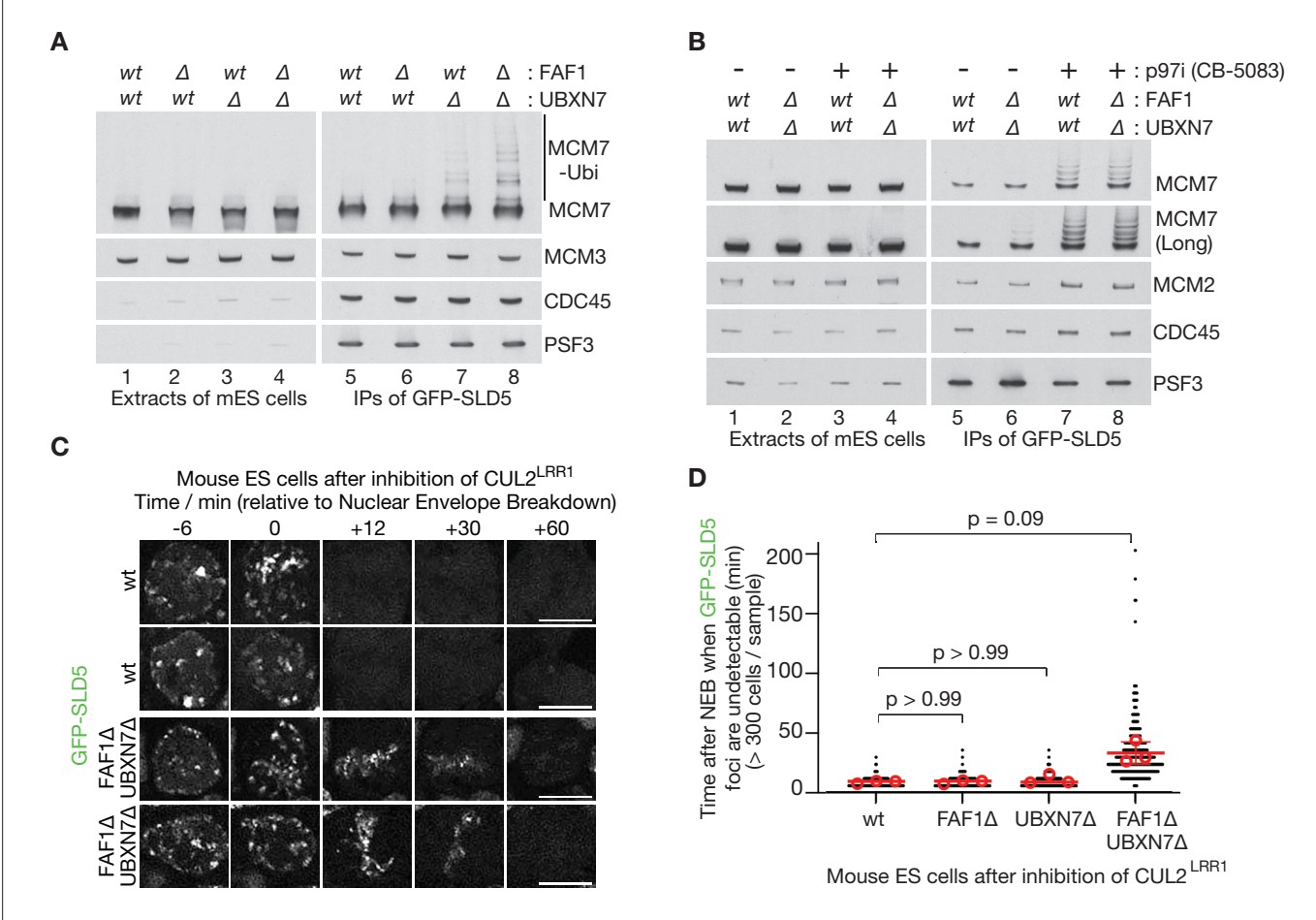

**Figure 6.** Partial redundancy between UBXN7 and FAF1 for CMG helicase disassembly during S-phase and mitosis in mouse ES cells. (**A**) CMG was isolated from extracts of mouse embryonic stem cells (mES) with the indicated genotypes, by immunoprecipitation of GFP-tagged SLD5 subunit of the helicase. (**B**) Equivalent experiment in which cells were treated as indicated with 5 µM CB-5083 (p97i = p97 inhibitor) for 3 hr before harvesting. (**C**) Time-lapse video analysis of mitotic entry in mouse ES cells expressing GFP-SLD5, following inhibition of CUL2$^{LRR1}$. Scale bars correspond to 10 µm. (**D**) Quantification of the data in (**C**). The individual data points from three independent experiments (>300 cells in total) are depicted as black dots in a scatter plot. The mean values from each experiment are shown in red circles, whilst red bars and error bars represent the average of the mean values and the associated standard deviations ($n = 3$). The samples were then compared by a Kruskal–Wallis test followed by Dunn's test, yielding the indicated p values. See also **Figure 6—figure supplements 3–5** (note that **Figure 6—figure supplement 4E** confirms that the difference between FAF1Δ UBXN7Δ and cells lacking either FAF1 or UBXN7 is statistically significant, despite the weaker significance of the data in (**D**)).

The online version of this article includes the following source data and figure supplement(s) for figure 6:

**Source data 1.** Source data for **Figure 6**.

**Figure supplement 1.** Sequence alignment of human and mouse orthologues of p97, NPL4, and UFD1.

**Figure supplement 2.** Sequence alignment of human and mouse orthologues of FAF1, FAF2, and UBXN7.

**Figure supplement 3.** The deletion of the *Faf1*, *Ubxn7*, and *Faf2* genes by CRISPR-Cas9.

**Figure supplement 3—source data 1.** Source data for **Figure 6—figure supplement 3**.

**Figure supplement 4.** Rescue of FAF1Δ UBXN7Δ mouse ES cells by expression of human FAF1 or UBXN7.

**Figure supplement 5.** Mitotic CMG disassembly is blocked in mES cells that lack TRAIP.

CRISPR-Cas9 was used to make deletions in exon 1 of the *Ubxn7* and *Faf1* genes (**Figure 6—figure supplement 3**), using mouse ES cells in which both copies of the *Gins4* (*Sld5*) gene had previously been modified to incorporate a GFP tag (**Villa et al., 2021**). The CMG helicase was then isolated from cell extracts, by immunoprecipitation of the GFP-tagged GINS4/SLD5 subunit. In extracts of control or FAF1Δ cells, CMG-MCM7 ubiquitylation was scarcely detectable (**Figure 6A**, lanes 5–6), reflecting the rapid and efficient disassembly of the ubiquitylated helicase by p97 (**Villa et al., 2021**). However,

ubiquitylated CMG-MCM7 was observed in extracts of UBXN7Δ cells (*Figure 6A*, lane 7) and accumulated slightly in cells that lacked both FAF1 and UBXN7 (*Figure 6A*, lane 8). The observed defects could be rescued by expression of full-length UBXN7 or FAF1 in UBXN7Δ FAF1Δ cells, though FAF1-ΔUBX or FAF1-ΔCoiled Coil were unable to rescue (*Figure 6—figure supplement 4A–D*). These data indicated that nuclear UBXN7 contributes to the disassembly of ubiquitylated CMG helicase during DNA replication termination in mouse ES cells, with a minor contribution from FAF1. Nevertheless, ubiquitylated CMG further accumulated when control or UBXN7Δ FAF1Δ cells were treated with p97 inhibitor (*Figure 6B*, lanes 5–8; the short ubiquitin chains on CMG-MCM7 under such conditions might reflect the reduction of free ubiquitin pools upon p97 inhibition), suggesting that CMG disassembly in the absence of UBXN7 and FAF1 was not blocked completely. It is possible that additional UBX proteins not characterised in this study contribute to CMG disassembly in the absence of UBXN7 and FAF1. Alternatively, or in addition, our data indicate that extensive ubiquitylation of CMG-MCM7 should bypass the requirement for UBX proteins (*Figure 1*).

A second pathway for the disassembly of ubiquitylated CMG helicase acts during mitosis in metazoan cells and is important to process sites of incomplete DNA replication (*Deng et al., 2019*; *Sonneville et al., 2019*; *Sonneville et al., 2017*; *Villa et al., 2021*). This pathway is independent of CUL2$^{LRR1}$ and instead requires the TRAIP ubiquitin ligase (*Deng et al., 2019*; *Sonneville et al., 2019*; *Sonneville et al., 2017*; *Villa et al., 2021*), which is mutated in human patients with a form of primordial dwarfism syndrome (*Harley et al., 2016*). To test whether FAF1 and UBXN7 also contribute to the mitotic disassembly of ubiquitylated CMG, we inactivated CUL2$^{LRR1}$ in mouse ES cells and monitored the presence on mitotic chromatin of GFP-SLD5. Time-lapse video analysis showed that CMG disassembly upon mitotic entry was blocked in the absence of TRAIP (*Figure 6—figure supplement 5*) as shown previously (*Villa et al., 2021*) and was slower in FAF1Δ UBXN7Δ compared to FAF1Δ, UBXN7Δ, or control cells (*Figure 6C, D*). This defect was efficiently rescued by expression of either UBXN7 or FAF1, but not by FAF1-ΔUBX or FAF1-ΔCC (*Figure 6—figure supplement 4E*). These data indicated that TRAIP-dependent CMG disassembly by p97 was delayed in cells that lacked both FAF1 and UBXN7, which act redundantly in the mitotic CMG disassembly pathway in mouse ES cells.

These findings suggested that the unfolding of other ubiquitylated substrates of p97-UFD1-NPL4 should also be impaired in the absence of FAF1 and UBXN7, particularly when ubiquitin chain formation on substrates of p97-UFD1-NPL4 is suboptimal. Cullin ligases represent the largest family of E3 enzymes directing K48-linked ubiquitin chain formation in eukaryotic cells (*Harper and Schulman, 2021*) and their activity is dependent upon neddylation of the cullin scaffold. Correspondingly, we found that FAF1Δ UBXN7Δ cells showed enhanced sensitivity to the neddylation inhibitor MLN4924 (*Soucy et al., 2009*), compared to control cells (*Figure 7A, B*), likely reflecting the defective processing of multiple targets of p97-UFD1-NPL4 in diverse aspects of cell biology. This defect was rescued by expression of wild type FAF1 protein but not FAF1-ΔUBX or FAF1-ΔCC (*Figure 7C*). Moreover, expression of UBXN7 restored the sensitivity of FAF1Δ UBXN7Δ cells to the level to FAF1Δ cells (compare *Figure 7B, D*). Overall, these findings suggest that the unfolding of ubiquitylated substrates by mammalian p97-UFD1-NPL4 is modulated both by the extent of ubiquitylation and by the action of multiple UBX proteins.

## Discussion

Together with previous studies of the conserved ubiquitin-binding modules of human UFD1-NPL4 (*Pan et al., 2021b*; *Pye et al., 2007*; *Ye et al., 2003*), our finding that the NPL4 groove is essential for CMG helicase disassembly in reconstituted reactions (*Figure 3—figure supplement 2*) indicates that human p97-UFD1-NPL4 initiates substrate unfolding by a fundamentally similar mechanism to yeast Cdc48-Ufd1-Npl4 (*Ji et al., 2022*; *Park et al., 2005*; *Twomey et al., 2019*). It is likely that the engagement of UFD1-NPL4 with K48-linked poly-ubiquitin leads to initial ubiquitin unfolding and translocation through the central pore of p97/Cdc48, followed by unfolding of the ubiquitylated polypeptide.

However, human p97-UFD1-NPL4 is dependent upon the conjugation of very long ubiquitin chains to the substrate, in the absence of other factors (*Figure 2*; *Figure 2—figure supplements 1 and 2*). Our data indicate that this high ubiquitin threshold is set by UFD1-NPL4 rather than by p97 itself (*Figure 3*). Although the NPL4-NZF domain is responsible for the majority of detectable ubiquitin binding by NPL4 in our assays (*Figure 3—figure supplement 2E*), as seen in past studies (*Meyer et al., 2002*; *Tsuchiya et al., 2017*), it is largely dispensable for CMG disassembly and thus for

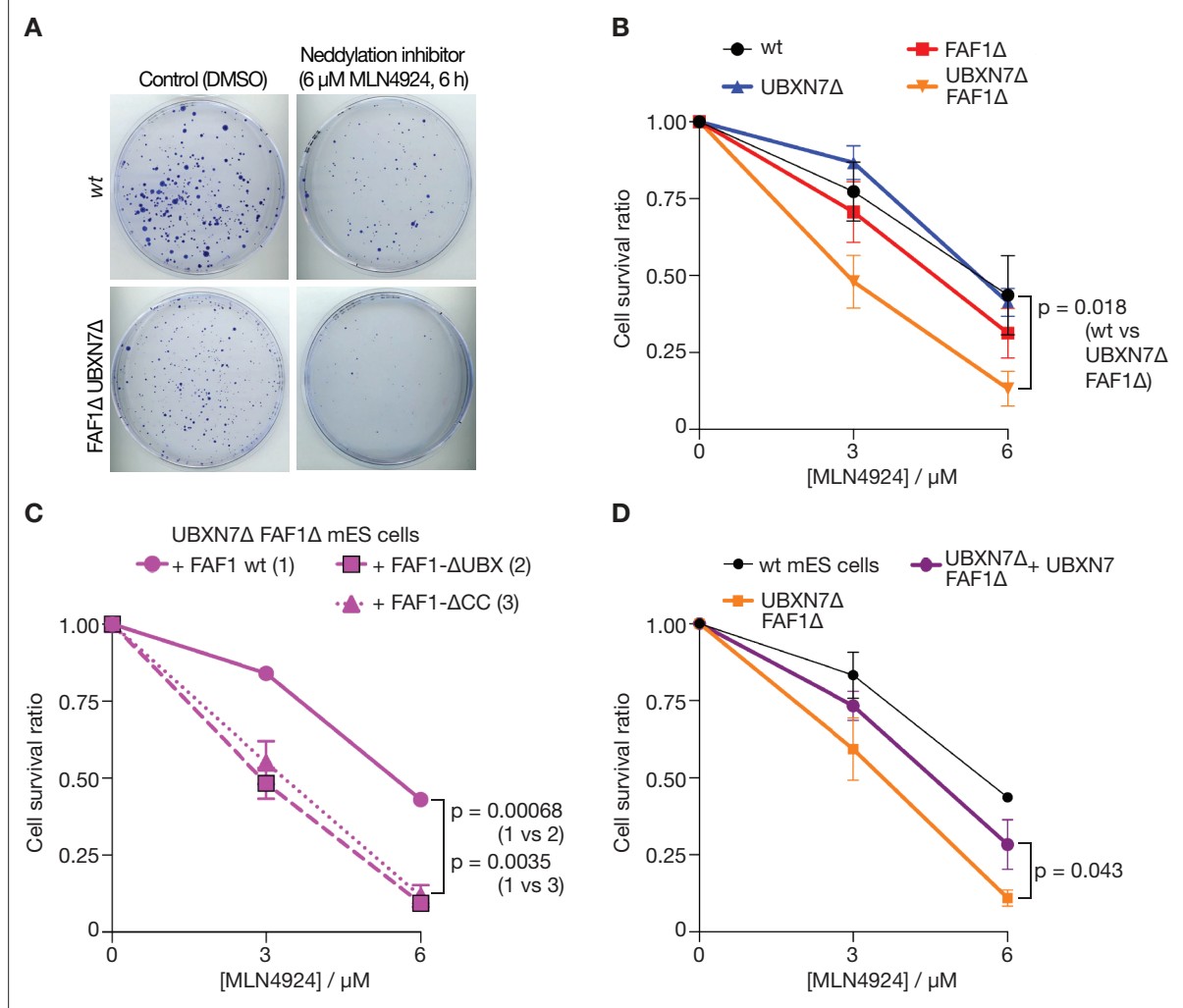

**Figure 7.** Cells lacking FAF1 and UBXN7 are sensitive to global inhibition of cullin ligase activity. (**A**) Cells of the indicated genotypes were treated as shown and then grown on 10 cm plates for 6 days. Surviving colonies were fixed and stained with crystal violet. (**B**) Viability of cells treated with 0, 3, or 6 μM MLN-4924 for 6 hr. The data represent the mean values and standard deviation for three independent experiments (*n* = 3). The mean survival ratios of wt and FAF1Δ UBXN7Δ after treatment with 6 μM MLN-4924 condition were compared via a two-tailed *t*-test. (**C**) Analogous experiment involving FAF1Δ UBXN7Δ mouse ES cells expressing the indicated versions of human FAF1 (as in *Figure 6—figure supplement 4B*) at the *Rosa26* locus. (**D**) Similar experiment to that in (**C**), comparing wt mouse ES cells, FAF1Δ UBXN7Δ cells, and FAF1Δ UBXN7Δ cells expressing human UBXN7 from the *Rosa26* locus.

substrate unfolding by p97-UFD1-NPL4 (*Figure 3—figure supplement 2C*), consistent with previous findings that the NPL4-NZF is not required for ERAD in human cells (*Ye et al., 2003*). These findings indicate that the dynamic binding to ubiquitylated substrate of conserved ubiquitin-binding modules within the NPL4 tower and the UFD1-UT3 domain (*Pan et al., 2021b*; *Pye et al., 2007*), favoured by the presence of very long K48-linked ubiquitin chains, is sufficient to initiate substrate unfolding by p97.

Our findings indicate that UBX proteins not only help to target p97 to substrates in a variety of subcellular locations but also play a direct role in the unfolding reaction, particularly when the ubiquitin chains conjugated to the substrate are short. Stimulation of p97-UFD1-NPL4 by UBX proteins is dependent not only on the presence of UFD1-NPL4 (*Figure 4*) but on conserved residues in the NPL4 groove (*Figure 4—figure supplement 2*, NPL4-AAE), indicating that substrate processing under such conditions is initiated by the trapping of an unfolded ubiquitin intermediate, as for yeast Cdc48-Ufd1-Npl4. UBXN7, FAF1, and FAF2 all bind directly to p97-UFD1-NPL4 (*Figure 4—figure supplement 1*; *Alexandru et al., 2008*), in each case using their UBX domain to associate with the

amino terminal face of the p97 hexamer (*Kim et al., 2011*; *Li et al., 2017*). We propose that these UBX proteins promote the formation of productive complexes between p97-UFD1-NPL4 and poly-ubiquitylated substrates, to facilitate the initiation of substrate unfolding. Consistent with this view, the ubiquitin threshold of human p97-UFD1-NPL4 in the presence of the UBX proteins is very similar to yeast Cdc48-Ufd1-Npl4 (*Figure 4E*), likely reflecting the engagement of the K48-linked ubiquitin chain by the multiple conserved ubiquitin-binding modules within human UFD1-NPL4.

In the case of UBXN7, stimulation of p97-UFD1-NPL4 in reconstituted assays is dependent not only on the UBX domain but also on the UBA and UIM motifs that bind ubiquitin (*Figure 5A–C*). This indicates that UBXN7 functions by binding simultaneously to p97-UFD1-NPL4 and the polyubiquitin chain on the substrate, thereby promoting the trapping of an unfolded ubiquitin intermediate on the NPL4 groove. It is possible that FAF1 and FAF2 function in an analogous manner, but we were unable to detect stable binding of the FAF1 coiled coil to K48-linked ubiquitin chains in vitro (*Figure 5— figure supplement 3*). However, the same is true for UFD1-NPL4-ΔNZF (*Figure 3—figure supplement 2E*), which nevertheless contains conserved low affinity ubiquitin-binding modules (*Pan et al., 2021b*; *Pye et al., 2007*) and supports the disassembly of ubiquitylated CMG in reconstituted assays (*Figure 3—figure supplement 2C*). Therefore, we cannot exclude that the FAF1 coiled coil also functions by binding to both p97-UFD1-NPL4 and ubiquitin. Structural studies of early intermediates in the unfolding reaction will be important in future studies, to elucidate further the underlying mechanism.

Our data indicate some degree of functional redundancy between UBXN7 and FAF1 in mammalian cells (*Figures 6 and 7*), which is likely to be mitigated by differences in their subcellular localisation (*Raman et al., 2015*), together with additional unique features of each factor such as the preferential association of the UBXN7-UIM domain with neddylated cullins (*den Besten et al., 2012*). The growth of UBXN7Δ FAF1Δ cells is further impaired in the absence of FAF2 (*Figure 6—figure supplement 3F*), which likely regulates p97-UFD1-NPL4 on the surface of the endoplasmic reticulum, lipid droplets, and the nuclear outer membrane (*Go et al., 2021*; *Raman et al., 2015*; *Zehmer et al., 2009*). The remaining activity of p97-UFD1-NPL4 in UBXN7Δ FAF1Δ FAF2Δ cells might involve additional UBX proteins that associate with p97-UFD1-NPL4 (*Raman et al., 2015*) and remain to be characterised in vitro. However, our findings also indicate (*Figure 1*) that substrates with very long ubiquitin chains can be unfolded by p97-UFD1-NPL4 in the absence of UBX proteins, consistent with previous in vitro assays with an artificial model substrate (*Blythe et al., 2017*). Nevertheless, *Ubxn7*, *Faf1*, and *Faf2* all become essential for viability during mouse development or soon after birth (*Adham et al., 2008*; *Koscielny et al., 2014*), indicating their importance for the biology of mammalian p97-UFD1-NPL4. Similarly, the *ubxn-3* orthologue of mammalian FAF1 is important for embryonic viability and adult fertility during *C. elegans* development (*Xia et al., 2021*). Our data suggest that UBX proteins such as UBXN7, FAF1, and FAF2 all increase the efficiency of substrate unfolding by p97-UFD1-NPL4, in diverse areas of eukaryotic cell biology. We suggest that such factors are particularly important whenever the length of the ubiquitin chain on the substrate is below the inherently high ubiquitin threshold of the mammalian unfoldase.

## Materials and methods

Reagents and resources used in this study, including all purified proteins, are listed in *Supplementary file 1*.

### Plasmid DNA construction

The plasmids generated in this study are shown in *Supplementary file 2*. PCRs were conducted with Phusion High-Fidelity DNA Polymerase (New England Biolabs, M0530), and amplified DNA fragments were cloned via the Gibson Assembly Cloning Kit (New England Biolabs, E2611) or with restriction enzymes and T4 ligase. All new constructs were verified by Sanger sequencing. Site-specific mutagenesis was performed using the Phusion polymerase, according to the manufacturer's protocol.

Coding sequences of human p97 (Uniprot identifier P55072-1), UFD1 (Uniprot identifier Q92890-2), NPL4 (Uniprot identifier Q8TAT6-1), FAF1 (Uniprot identifier Q9UNN5-1), and UBXN7 (Uniprot identifier O94888-1) were amplified from XpressRef Universal Total human RNA (QIAGEN, 338112) by RT-PCR (TaKaRa, RR014) and cloned by Gibson assembly into the vector pK27SUMO (expressing a 14His-tagged version of the yeast SUMO protein Smt3) or pET28c. The coding sequences of human

UBXN1 (Uniprot identifier Q04323-1) and FAF2 (Uniprot identifier Q96CS3-1) were synthesised by GenScript Biotech. For FAF2, the residues encoding the membrane anchoring domain (amino acids 90–118) were removed by PCR to improve solubility of the expressed protein.

## Expression of proteins in bacterial cells

Each plasmid was transformed into Rosetta (DE3) pLysS (Novagen, 70956), which was grown in LB medium supplemented with kanamycin (50 µg/ml) and chloramphenicol (35 µg/ml). Subsequently, a 100-ml culture was grown overnight at 37°C with shaking at 200 rpm. The following morning, the culture was diluted fivefold with 400 ml of LB medium supplemented with kanamycin (50 µg/ml) and chloramphenicol (35 µg/ml) and then left to grow at 37°C until an OD600 of 0.8 was reached. Protein expression was then induced overnight at 18°C in the presence of 1 mM IPTG (Isopropyl β-d-1-thiogalactopyranoside). Cells were harvested by centrifugation for 15 min in a JLA-9.1000 rotor (Beckman) at $5180 \times g$ and the pellets were stored at −20°C.

## Protease inhibitors

One tablet of 'Roche cOmplete EDTA-free protease inhibitor Cocktail' (Roche, 11873580001) was either added directly to 20 ml of protein purification buffer or else dissolved in 1 ml of water to make 20× stock (EDTA = ethylenediaminetetraacetic acid).

## Buffers

Lysis buffer: 50 mM Tris–HCl (pH 8.0), 0.5 M NaCl, 5 mM Mg(OAc)$_2$, and 0.5 mM TCEP (Tris Carboxy Ethyl Phosphene).

Gel filtration buffer: 20 mM HEPES–KOH (pH 7.9; HEPES = (4-(2-hydroxyethyl)-1-piperazineethane sulfonic acid)), 0.15 M NaCl, 5 mM Mg(OAc)$_2$, 0.3 M sorbitol, and 0.5 mM TCEP.

Reaction buffer: 25 mM HEPES–KOH (pH 7.9), 10 mM Mg(OAc)$_2$, 0.02% (wt/vol) IGEPAL CA-630, 0.1 mg/ml bovine serum albumin (BSA), and 1 mM DTT (dithiothreitol).

Wash buffer: 100 mM HEPES–KOH (pH 7.9), 100 mM KOAc, 10 mM Mg(OAc)$_2$, 2 mM EDTA, 0.1% (wt/vol) IGEPAL CA-630.

Cell Lysis buffer: 100 mM HEPES–KOH (pH 7.9), 100 mM KOAc, 10 mM Mg(OAc)$_2$, 2 mM EDTA, 10% (wt/vol) glycerol, 0.1% (wt/vol) Triton X-100, 2 mM sodium fluoride, 2 mM sodium β-glycerophosphate pentahydrate, 10 mM sodium pyrophosphate, 1 mM sodium orthovanadate, 1 µg/ml LR-microcystin, 1 mM DTT, and Roche cOmplete EDTA-free protease inhibitor Cocktail.

## Purification of human p97 and its ATPase mutants

Bacterial pellets expressing [14His-Smt3]-p97 from 500 or 1000 ml cultures were resuspended in 20 ml Lysis buffer supplemented with 40 mM imidazole, 0.1 mM ATP, Roche cOmplete EDTA-free protease inhibitor Cocktail. The samples were lysed by incubation with 1 mg/ml lysozyme on ice for 30 min, followed by sonication for 90 s (15 s on, 30 s off) at 40% on a Branson Digital Sonifier, then clarified by centrifugation at $10,000 \times g$ for 30 min in an SS-34 rotor (Sorvall).

The supernatant was mixed with 1 ml resin volume of Ni-NTA beads (Qiagen, 30210) which was pre-equilibrated with Lysis buffer supplemented with 40 mM imidazole and 0.1 mM ATP. After 1-hr incubation at 4°C, beads were recovered in a disposable gravity flow column and washed with 150 ml of Lysis buffer containing 40 mM imidazole and 0.1 mM ATP, followed by 10 ml of Lysis buffer containing 60 mM imidazole, 2 mM ATP, and 10 mM MgOAc. [14His-Smt3]p97 was eluted with 5 ml of Lysis buffer containing 400 mM imidazole and 0.1 mM ATP. Ulp1 protease (10 µg/ml) was added to the eluate and incubated at 4°C overnight, to cleave the 14His-Smt3 tag from p97. The sample was then loaded onto a 24 ml Superose 6 column equilibrated in Gel filtration buffer containing 0.1 mM ATP. Fractions containing hexameric p97 were pooled, concentrated with an Amicon Ultra-4 Centrifugal Filter Unit (Merck, UFC803024) according to the manufacture's protocol, aliquoted and snap-frozen with liquid nitrogen.

## Purification of human UFD1-NPL4

Human untagged NPL4 (or [6His]NPL4) and [14His-Smt3]UFD1 were expressed individually in 750 and 250 ml cultures (at a 3:1 ratio of NPL4 to UFD1). The respective bacterial pellets were then resuspended in 15 and 5 ml of Lysis buffer containing 30 mM imidazole, Roche protease inhibitor tablets. The samples

were then mixed and lysed by incubation with 1 mg/ml lysozyme on ice for 30 min, followed by sonication for 90 s (15 s on, 30 s off) at 40% on a Branson Digital Sonifier, before clarification by centrifugation at 10,000 × *g* for 30 min in an SS-34 rotor.

The supernatant was mixed with 1 ml resin volume of Ni-NTA beads which had been pre-equilibrated with Lysis buffer containing 30 mM imidazole. After 1-hr incubation at 4°C, beads were recovered in a disposable gravity flow column and washed with 150 ml of Lysis buffer containing 30 mM imidazole, followed by 10 ml of Lysis buffer containing 50 mM imidazole, 2 mM ATP, and 10 mM MgOAc. The complex of 14His-Smt3UFD1 with NPL4 was eluted with 5 ml of Lysis buffer containing 400 mM imidazole.

The yeast Ulp1 protease (10 µg/ml) was then added to cleave the 14His-Smt3 tag from UFD1, and the mixture was dialyzed into 500 ml of Lysis buffer containing 30 mM imidazole at 4°C overnight. The dialyzed mixture was mixed with 0.5 ml Ni-NTA beads equilibrated with Lysis buffer containing 30 mM imidazole. After 15 min of rotation at 4°C, the flow-through fraction was collected and concentrated with an Amicon Ultra-4 Centrifugal Filter Unit according to the manufacture's protocol and then loaded onto a 24-ml Superdex 200 column equilibrated in Gel filtration buffer. Fractions containing UFD1-NPL4 were pooled, concentrated, aliquoted, and snap-frozen with liquid nitrogen.

For purification of 14His-Smt3UFD1-NPL4 proteins in *Figure 3—figure supplement 2*, the tag cleavage by Ulp1 and dialysis steps were omitted and purified similarly.

## Purification of metazoan UBX proteins

Bacterial pellets expressing 14His-Smt3-tagged UBX proteins from 250 to 1000 ml cultures were resuspended in 20 ml of Lysis buffer containing 30 mM imidazole and Roche cOmplete EDTA-free protease inhibitor Cocktail. The samples were lysed by incubation with 1 mg/ml lysozyme on ice for 30 min, followed by sonication for 90 s (15 s on, 30 s off) at 40% on a Branson Digital Sonifier, before clarification by centrifugation at 10,000 × *g* for 30 min in an SS-34 rotor.

The supernatant was mixed with 1 ml resin volume of Ni-NTA beads which was pre-equilibrated with Lysis buffer containing 30 mM imidazole. After 1-hr incubation at 4°C, beads were recovered in a disposable gravity flow column and washed with 150 ml of Lysis buffer containing 30 mM imidazole, followed by 10 ml of Lysis buffer containing 50 mM imidazole, 2 mM ATP, and 10 mM MgOAc. 14His-Smt3-tagged UBX proteins were eluted with 5 ml of Lysis buffer containing 400 mM imidazole. Ulp1 protease (10 µg/ml) was then added to cleave the 14His-Smt3 tag from UBX proteins during an overnight incubation. Subsequently, the samples were concentrated with an Amicon Ultra-4 Centrifugal Filter Unit according to the manufacture's protocol and loaded onto a 24-ml Superdex 200 column in Gel filtration buffer. In the case of the fragments comprising the UBX domain of FAF1 and UBXN-3, a Superdex75 column was used instead of Superdex 200. The peak fractions containing each UBX protein were pooled, aliquoted, and snap-frozen with liquid nitrogen.

For purification of 14His-Smt3FAF1 proteins shown in *Figure 5—figure supplement 3*, eluted proteins from the Ni-NTA beads were dialysed into Gel filtration buffer, and snap-frozen with liquid nitrogen.

## Generation of a budding yeast strain expressing CMG-Mcm7-KR

A yeast strain coexpressing the 11 subunits of budding yeast CMG was generated as described previously (*Deegan et al., 2020*; *Zhou et al., 2017*), with the following modification. The original plasmid expressing *MCM7* and *MCM6* genes from the bi-directional *GAL1,10* promoter at the *leu2* locus (pJF4, *Frigola et al., 2013*) was replaced with a modified version, which introduced mutations in surface lysines of Mcm7 (pCPR037, see *Supplementary file 1*, *Supplementary file 2*) that restricted CMG ubiquitylation by SCF^Dia2 to a single chain per CMG-Mcm7 subunit (*Figure 1—figure supplement 1C*). In the resulting strain (*yCPR337*, see *Supplementary file 1*) protein expression for each CMG subunit was monitored by sodium dodecyl sulfate polyacrylamide gel electrophoresis (SDS–PAGE) and immunoblotting.

## Purification of wild-type yeast CMG and CMG-Mcm7-KR

Budding yeast cells (yTDK20 for wt CMG, yCPR337 for CMG-Mcm7-KR) were grown at 30°C in rich medium (1% [wt/vol] yeast extract, 212750, Becton Dickinson and 2% [wt/vol] bacteriological peptone, LP0037B, Oxoid) supplemented with 2% raffinose. Once the cultures reached a cell density of 2–3 × 10^7 cells/ml, galactose was added to 2% to induce expression of CMG subunits from the *GAL1,10* promoter, for 3 hr at 30°C. Subsequently, cells were collected by centrifugation at 3315 × *g*

for 10 min and washed once in buffer containing 25 mM HEPES–KOH (pH 7.6), 10% glycerol, 0.02% Tween-20, 2 mM MgOAc, 1 mM DTT, and 300 mM KCl (CMG buffer/300 mM KCl). After the wash, cell pellets were resuspended in 0.3 volumes of CMG buffer/300 mM KCl supplemented with 1× Complete Protease Inhibitor Cocktail (11873580001, Roche; one tablet dissolved in 1 ml water makes 25× stock solution). The cell suspension was then frozen dropwise in liquid nitrogen to make 'yeast popcorn'. Subsequently, the frozen yeast popcorn was ground in a freezer mill (SPEX CertiPrep 6850 Freezer Mill) using four cycles of 2 min at a rate of 15 cycles per second. The resulting powder was stored in −80°C until required.

Thawed yeast cell powder was resuspended with an equivalent volume of CMG buffer/300 mM KCl containing protease inhibitors and the sample was centrifuged at 235,000 × *g* at 4°C for 1 hr. The soluble extract was recovered and mixed with 3 ml of anti-FLAG M2 affinity resin (A2220, Sigma), to bind CMG via an internal FLAG tag on the Cdc45 subunit, and the mixture was then incubated at 4°C for 2 hr on a rotating wheel. The resin was then recovered and washed extensively with 200 ml of CMG buffer/300 mM KCl lacking protease inhibitors. Elution was then performed by mixing the beads with 5 ml of CMG buffer/300 mM KCl containing 0.5 mg/ml 3XFLAG peptide (F4799, Sigma), and with another 5 ml of CMG buffer/300 mM KCl containing 0.25 mg/ml 3XFLAG peptide. The recovered 10 ml eluate was supplemented with 2 mM $CaCl_2$ and mixed with 1 ml Calmodulin Sepharose 4B (7052901, Cytiva) for 1 hr at 4°C to capture purified CMG via CBP-tagged Mcm3 subunit. The resin was then collected and washed extensively with 100 ml of CMG buffer/300 mM KCl supplemented with 2 mM $CaCl_2$ and Roche protease inhibitors. CMG complexes were then eluted in 15 ml of CMG buffer/300 mM KCl containing 2 mM EDTA and 2 mM EGTA (Ethyleneglycol Bis(2-Aminoethyl Ether)-N,N,N',N' Tetraacetic Acid). Eluate fractions were loaded onto a 0.24 ml MiniQ column equilibrated in CMG buffer/300 mM KCl and the helicase complex was eluted using a 4 ml gradient ranging from 300 to 600 mM KCl in CMG buffer. Fractions containing CMG complexes were pooled and dialysed against CMG buffer/200 mM KOAc for 4 hr at 4°C, recovered, aliquoted and snap frozen, before storage at −80°C.

## Purification of budding yeast proteins

The budding yeast proteins used in this study (replisome factors, ubiquitylation factors, and Cdc48 plus its Ufd1-Npl4 adaptors) are described in *Supplementary file 1*. The various factors were expressed using plasmids and strains described in *Supplementary file 1* and were purified as described previously (*Deegan et al., 2020*).

## Purification of *C. elegans* proteins

The *C. elegans* proteins used in this study (replisome factors, ubiquitylation factors, neddylation factors, and CDC-48 plus adaptors) are described in *Supplementary file 1*. The various factors were expressed using plasmids and strains described in *Supplementary file 1* and were purified as described previously (*Xia et al., 2021*), except for UBXN-3-Δ435 and UBXN-3-Δ527 that are depicted in *Figure 5—figure supplement 2* and were expressed as fusions to 14His-Smt3 in *E. coli* using the plasmids pRF056 and pRF057, as described above for other UBX proteins.

## Preparation of IR Dye-labelled K48-linked ubiquitin chains

Fluorescently labelled K48-linked ubiquitin chains were synthesised as described previously (*Abdul Rehman et al., 2021*; *Abdul Rehman et al., 2016*) with slight modifications. K48-linked ubiquitin chains were synthesised in a reaction mix containing 2500 µM wt ubiquitin, 250 µM [6His]-ubiquitin (K48R, K63C), 0.5 µM UBE1, 15 µM UBE2R1, 10 mM ATP, 50 mM Tris–HCl (pH 7.5), 10 mM $MgCl_2$, and 0.6 mM DTT with overnight incubation at 30°C. Chains with successful incorporation of [6His]-ubiquitin (K48R, K63C) at their distal end were purified using a HisTrap FF 5 ml column (17525501, Cytiva). Subsequently, ubiquitin chains were fractionated using a Superdex 200 16/60 column. Fractions containing chains with 7–20 ubiquitins were pooled and concentrated, before buffer exchange into phosphate-buffered saline (PBS). The purified ubiquitin chains were then reacted with a threefold molar excess of IRDye-800CW Maleimide (929-80020, Li-Cor) for 3 hr at 22°C with gentle agitation. The reaction was quenched with 50 mM β-mercaptoethanol. Subsequently, 500 µl of the labelled ubiquitin chains (5.5 mg/ml) were incubated with 20 µl of Precission protease (2.5 mg/ml) at 4°C overnight to cleave the 6His-tag off ubiquitin, then incubated with 500 µl slurry of Ni-NTA beads at 4°C for 30 min. The

unbound fraction was then concentrated and buffer exchanged into 50 mM Tris–HCl (pH 7.5), before being aliquoted and snap-frozen with liquid nitrogen.

## Reconstituted ubiquitylation and disassembly of yeast CMG helicase

Yeast CMG was ubiquitylated in reconstituted in vitro reactions as described previously (*Deegan et al., 2020*). Reactions were set up using 'Reaction buffer' containing 150 mM KOAc and 2 mM ATP, in a final volume of 10 µl and a final concentration of 15 nM yeast CMG, 30 nM Uba1, the concentrations of the E2 Cdc34 and E3 SCF$^{Dia2}$ that are discussed below, 30 nM Ctf4 and 6 µM ubiquitin. Concentrations of E2 and E3 were as specified in the figures and figure legends. Ubiquitylation reactions were conducted at 30°C for 20 min and then stopped by the addition of KOAc to a final concentration of 700 mM.

The ubiquitylated yeast CMG was then incubated for 30 min at 4°C with 2.5 µl (per 10 µl ubiquitylation reaction mix) magnetic beads (Dynabeads M-270 Epoxy; Life Technologies, 14302D) that had been conjugated to anti-FLAG M2 antibodies. After the incubation, the bead-bound protein complexes were washed twice with 190 µl of Reaction buffer containing 100 mM KOAc and then resuspended in Reaction buffer containing 100 mM KOAc and 5 mM ATP. For the disassembly reactions, 50 nM each of human p97 (or yeast Cdc48) hexamers, human UFD1-NPL4 (or yeast Ufd1-Npl4), and the indicated human UBX proteins were added to the suspension of beads bearing ubiquitylated yeast CMG, before incubation at 30°C for 20 min with shaking at 1000 rpm on an Eppendorf ThermoMixer. The supernatants were collected and the beads washed twice with 190 µl of Reaction buffer containing 700 mM KOAc, before elution of bound proteins by the addition of LDS-PAGE sample loading buffer (Invitrogen, NP0007; LDS = lithium dodecyl sulphate) and heating for 3 min at 95°C.

## Reconstituted ubiquitylation and disassembly of *C. elegans* CMG helicase

For the experiment in *Figure 5—figure supplement 2B*, worm CMG was ubiquitylated in reconstituted in vitro reactions as described previously (*Xia et al., 2021*), using proteins described in *Supplementary file 1*. Reactions were set up using 'Reaction buffer' containing 100 mM KOAc and 5 mM ATP, in a final volume of 10 µl and a final concentration of 15 nM worm CMG, 50 nM UBA-1, 300 nM LET-70, 300 nM UBC-3, 50 nM ULA-1_RFL-1, 300 nM UBC-12, 100 nM DCN-1, 500 nM NED-8, 15 nM CUL-2$^{LRR-1}$, 60 nM TIM-1_TIPIN-1, 30 nM POLε, 30 nM CLSP-1, 30 nM MCM-10, 60 nM CTF-18_RFC, and 20 nM CTF-4. Ubiquitylation reactions were conducted at 20°C for 20 min.

The ubiquitylated worm CMG was then incubated for 30 min at 4°C with 2 µl (per 10 µl ubiquitylation reaction mix) magnetic beads (Dynabeads M-270 Epoxy; Life Technologies, 14302D) that had been coupled to anti-worm SLD-5 antibodies. After the incubation, the bead-bound protein complexes were washed twice with 190 µl of Reaction buffer containing 100 mM KOAc and then resuspended in Reaction buffer containing 5 mM ATP. For the disassembly reactions, 200 nM of *C. elegans* CDC-48.1 hexamer, 50 nM of *C. elegans* UFD-1_NPL-4.1 and 50 nM of the indicated form of *C. elegans* UBXN-3 were added to the suspension of beads bearing ubiquitylated worm CMG, in a final volume of 10 µl. The reactions were then incubated at 20°C for 20 min, whilst shaking at 1000 rpm on an Eppendorf ThermoMixer. The supernatants were then collected and the beads washed twice with 190 µl of Reaction buffer containing 700 mM KOAc. The bound proteins were eluted by addition of LDS-PAGE sample loading buffer (Invitrogen, NP0007) and heating for 3 min at 95°C.

## Monitoring the interaction of UFD1-NPL4 with p97

For the experiment in *Figure 3—figure supplement 2*, 3 µg of $^{14His-Smt3}$UFD1-NPL4 and 13 µg of p97 were mixed as indicated with 2 µl slurry of Ni-NTA beads in a total of 15 µl using 'Gel filtration buffer' containing 30 mM imidazole and 2 mM ATP. After 15-min incubation on ice, the Ni-NTA beads were washed four times with 1 ml of 'Wash buffer' containing 20 mM imidazole and 0.5 mM ATP. Bead-bound proteins were eluted in LDS-PAGE sample loading buffer containing 50 mM EDTA, by heating for 3 min at 95°C.

## Monitoring the interaction of UBX proteins with p97-UFD1-NPL4

For the experiments in *Figure 4—figure supplement 1* and *Figure 5—figure supplement 1*, 3 µg of $^{6His}$NPL4-UFD1, 3 µg of each UBX protein, and 13 µg of p97 were mixed as indicated with 2 µl slurry

of Ni-NTA beads in a total of 11 µl, using 'Gel filtration buffer' containing 30 mM imidazole and 2 mM ATP. After 15-min incubation on ice, the Ni-NTA beads were washed three times with 200 µl of 'Wash buffer' containing 20 mM imidazole and 0.5 mM ATP. Bead-bound proteins were eluted in LDS-PAGE sample loading buffer containing 50 mM EDTA, by heating for 3 min at 95°C.

## Monitoring the interaction of UFD1-NPL4 or FAF1 with K48-linked ubiquitin chains

For the experiments in *Figure 3—figure supplement 2E* and *Figure 5—figure supplement 3*, 15 µg of $^{14His-Smt3}$UFD1-NPL4 or 15 µg of $^{14His-Smt3}$FAF1, plus 2 µg of labelled K48-linked ubiquitin chains were mixed with 5 µl slurry of Ni-NTA beads, in a total of 15 µl using 'Gel filtration buffer' containing 30 mM imidazole. After 15-min incubation on ice, the Ni-NTA beads were washed four times with 1 ml of 'Wash buffer' containing 20 mM imidazole. Bead-bound proteins were eluted in LDS-PAGE sample loading buffer containing 50 mM EDTA, by heating for 3 min at 95°C. After SDS–PAGE, gels were stained with InstantBlue (Expedeon, ISB1L) and scanned with an Odyssey CLx imaging System (Li-Cor) to detect the labelled ubiquitin chains.

## SDS–PAGE and immunoblotting

Protein samples were resolved by SDS–PAGE in NuPAGE Novex 4–12% Bis-Tris gels (Thermo Fisher Scientific, NP0321 and WG1402A) either with NuPAGE MOPS SDS buffer (Thermo Fisher Scientific, NP0001) or NuPAGE MES SDS buffer (Thermo Fisher Scientific, NP0002), or with NuPAGE Novex 3–8% Tris–Acetate gels (Thermo Fisher Scientific, EA0375BOX and WG1602BOX) using NuPAGE Tris-Acetate SDS buffer (Thermo Fisher Scientific, LA0041). Resolved proteins were either stained with InstantBlue (Expedeon, ISB1L) or were transferred to a nitrocellulose membrane with the iBlot2 Dry Transfer System (Invitrogen, IB21001S). Antibodies used for protein detection in this study are described in *Supplementary file 1*. Conjugates to horseradish peroxidase of anti-sheep IgG from donkey (Sigma-Aldrich, A3415), or anti-mouse IgG from goat (Sigma-Aldrich, A4416) were used as secondary antibodies before the detection of chemiluminescent signals on Hyperfilm ECL (cytiva, 28906837, 28906839) with ECL Western Blotting Detection Reagent (cytiva, RPN2124).

## Preparation of antibody-coated magnetic beads

Firstly, 300 mg of Dynabeads M-270 Epoxy (Thermo Fisher Scientific, 14302D) were resuspended in 10 ml of dimethylformamide. Subsequently, 425 µl slurry of activated magnetic beads, which corresponded to ~1.4 × 10$^9$ beads, were washed twice with 1 ml of 1 M NaPO$_3$ (pH 7.4). The beads were then incubated with 300 µl of 3 M (NH4)$_2$SO$_4$, 300 µg of anti-FLAG M2 antibody (Sigma-Aldrich, F3165), and 1 M NaPO$_3$ (pH 7.4) up to a total volume of 900 µl. The mixture was then incubated at 4°C for 2 days with rotation. Finally, the beads were treated as follows: four washes with 1 ml PBS; 10 min in 1 ml PBS containing 0.5 % (wt/vol) IGEPAL CA-630 with rotation at room temperature; 5 min in 1 ml PBS containing 5 mg/ml BSA with rotation at room temperature; one wash with 1 ml PBS containing 5 mg/ml BSA. The beads were then resuspended with 900 µl PBS containing 5 mg/ml BSA and stored at 4°C.

## Cell lines

This study utilised E14tg2A mouse cells that were described previously (*Villa et al., 2021*) and validated by DNA sequencing of genomic loci.

## Growth of mouse ES cells

Mouse ES cells were grown as described previously (*Villa et al., 2021*). E14tg2a cells (*Supplementary file 1*) were cultured in a humidified atmosphere of 5% CO$_2$, 95% air at 37°C under feeder-free conditions with Leukemia Inhibitory Factor (LIF; MRC PPU Reagents and Services DU1715) in serum-containing medium. All culturing dishes were precoated with PBS containing 0.1% (wt/wt) gelatin (Sigma-Aldrich, G1890) for at least 5 min prior to seeding. The medium was based on Dulbecco's modified Eagle medium (DMEM; Thermo Fisher Scientific, 11960044), supplemented with 10% fetal bovine serum (FCS-SA/500, Labtech), 5% knockout serum replacement (Thermo Fisher Scientific, 10828028), 2 mM L-glutamine (Thermo Fisher Scientific, 25030081), 100 U/ml penicillin–streptomycin (Thermo Fisher Scientific, 15140122), 1 mM sodium pyruvate (Thermo Fisher Scientific, 11360070), a

mixture of seven non-essential amino acids (Thermo Fisher Scientific, 11140050), 0.05 mM β-mercaptoethanol (Sigma-Aldrich, M6250) and 0.1 µg/ml LIF. For passaging, cells were released from dishes using 0.05% trypsin–EDTA (Thermo Fisher Scientific, 25300054).

To determine doubling times, 200,000 or 500,000 cells (initial cell number N1) were grown for 48 hr on a 6-well plate, before recovery with 0.05% trypsin/EDTA. The total cell number in the final suspension (final cell number N2) was determined using a CellDrop BF (DeNovix). The doubling time (G) was then calculated via the formula: $G = (48 \times \log(2))/(\log(N2) - \log(N1))$. The experiments were performed in triplicate and mean values were determined, together with the standard deviation (SD).

## CRISPR-Cas9 genome editing

To design a pair of guide RNAs (gRNAs) to target a specific site in the mouse genome, we used 'The CRISPR Finder' provided by the Welcome Sanger Institute (https://wge.stemcell.sanger.ac.uk//find_crisprs). Annealed oligonucleotides containing the homology region were phosphorylated with T4 polynucleotide kinase (New England Biolabs, M201) and then ligated into the Bbs1 site of the vectors pX335 and pKN7 (*Supplementary file 1*) in the presence of T4 DNA ligase (New England Biolabs, M202), as previously described (*Pyzocha et al., 2014*).

To create small deletions at a particular locus, mouse ES cells were transfected with two plasmids expressing the chosen pair of gRNAs together with the Cas9-D10A 'nickase' mutant and the Puromycin resistance gene, as described below. FAF1Δ UBXN7Δ cells were produced by targeting the *Ubxn7* locus in FAF1Δ cells. Similarly, FAF1Δ UBXN7Δ FAF2-ΔUBX cells were produced by targeting the *Faf2* locus in FAF1Δ UBXN7Δ cells.

To express human FAF1 or UBXN7 from the CAG promoter in FAF1Δ UBXN7Δ mouse ES cells, we made donor DNA constructs in which the *CAG promoter-FAF1* or the *CAG promoter-UBXN7* were flanked with 0.5–1 kb of homology to either side of the *Rosa26* target locus. FAF1Δ UBXN7Δ cells were then transfected with 1.5 µg donor plasmid DNA together with 1 µg each of the plasmid pKN88 that expresses Cas9 nickase, a gRNA targeting *Rosa26* locus and the Puromycin resistance marker.

## Transfection of mouse ES cells and selection of clones

A stock solution of 1 mg/ml linear polyethylenimine (PEI; Polysciences, Inc, 24765-2) was prepared in a buffer containing 25 mM HEPES, 140 mM NaCl, 1.5 mM $Na_2HPO_4$ adjusted to pH 7.0 with NaOH. The solution was sterilised by passing through a 0.2-µm filter and stored at −20°C. 1 µg of each plasmid DNA (two plasmids expressing Cas9, gRNAs and the Puromycin resistance gene) were mixed in 100 µl of reduced-serum medium OPTI-MEM (Thermo Fisher Scientific, 31985062) then mixed with 15 µl of 1 mg/ml PEI, vortex well and left for 15 min. Subsequently, $1.0 \times 10^6$ cells were aliquoted and centrifuged at 350 × g for 5 min before resuspension in 200 µl OPTI-MEM. The cell suspension and DNA-PEI mix were gently mixed and incubated at room temperature for 30 min in a 1.5 ml microfuge tube. Cells were then transferred to a single well of a 6-well plate that contained 2 ml of complete DMEM medium. Cells were incubated for 24 hr after transfection, followed by two 24-hr rounds of selection with fresh medium containing 2 µg/ml Puromycin (Thermo Fisher Scientific, A1113802). The surviving cells were then released from the wells with 0.05% trypsin/EDTA, diluted 5/25/125 times with complete DMEM medium and then plated on 10 cm plates precoated with PBS containing 0.1% (wt/wt) gelatin. Subsequently, colonies were expanded and monitored as appropriate by immunoblotting, PCR, and DNA sequencing of the target locus.

## Genotyping of mouse ES cells by PCR and DNA sequencing

Cells from a single well of a 6-well plate were resuspended in 100 µl of 50 mM NaOH and the sample was then heated at 95°C for 15 min. Subsequently, 11 µl of 1 M Tris–HCl (pH 6.8) was added to neutralise the pH. A 0.5 µl aliquot of the resulting genomic DNA solution was used as the template for genotyping purposes in 25 µl PCR reactions with Ex Taq DNA polymerase (TaKaRa, RR001). PCR products were subcloned with the TOPO TA cloning Kit (Invitrogen, K457540) and sequenced with T3 primer (5'-AATTAACCCTCACTAAAGGG-3').

## Extracts of mouse ES cells and immunoprecipitation of protein complexes

$1.2 \times 10^7$ mouse ES cells were plated on a 15-cm Petri dish and grown for 24 hr at 37°C. Cells were released from the dishes by incubation for 10 min with 10 ml of PBS containing 1 mM EGTA, 1 mM EDTA and then harvested by centrifugation at $350 \times g$ for 3 min, before snap-freezing with liquid nitrogen and storage at −80°C. Cell pellets from two 15-cm Petri dishes were resuspended with an equal volume of 'Cell Lysis buffer' supplemented with 5 µM Propargyl-Ubiquitin (MRC PPU Reagents and Services, DU49003) to inhibit deubiquitylase activity, and chromosomal DNA was then digested for 30 min at 4°C with 1600 U/ml of Pierce Universal Nuclease (Thermo Fisher Scientific, 88702). The extracts were centrifuged at $20,000 \times g$ for 30 min at 4°C. The supernatant was then mixed with 100 µl slurry of Protein A/G Sepharose beads (Expedeon, AGA1000) that had been pre-washed in cell Lysis buffer. This step was used to remove proteins that bound non-specifically to the beads. After 15 min of rotation at 4°C, the samples were centrifuged at $800 \times g$ for 30 s at 4°C and the supernatant was recovered. A 20 µl aliquot of the supernatant was added to 40 µl of 1.5 × LDS sample loading buffer.

The rest of the extract was then incubated for 90 min with 10 µl slurry of GFP-Trap Agarose beads (gta-100, Chromotek). The beads were washed four times with 1 ml of Wash buffer and bound proteins were eluted at 95°C for 3 min in 30 µl of 1× LDS sample loading buffer.

## Inactivation of CUL2^LRR1 in mouse ES cells

Lipofectamine RNAiMAX Transfection Reagent (Thermo Fisher Scientific, 13778075) was used to introduce siRNA into mouse ES cells, according to the manufacturer's protocol. *LRR1* siRNA (Horizon Discovery, J-057816-10) were transfected at a final concentration of 25 nM. After 24 hr, the transfection was repeated and incubation continued for a further 24 hr. Subsequently, cells were treated with 5 µM MLN-4924 (Activebiochem, A1139) for 5 hr before imaging. MLN-4924 inhibits the E1 enzyme of the protein neddylation pathway and thereby inhibits cullin ligase activity (*Soucy et al., 2009*).

## Analysis of mitotic CMG helicase disassembly by spinning disk confocal microscopy

Confocal images of live cells were acquired with a Zeiss Cell Observer SD microscope with a Yokogawa CSU-X1 spinning disk, using a HAMAMATSU C13440 camera with a PECON incubator, a 60 × 1.4 NA Plan-Apochromat oil immersion objective, and appropriate excitation and emission filter sets. Images were acquired using the 'ZEN blue' software (Zeiss) and processed with ImageJ software (National Institutes of Health) as previously described (*Sonneville et al., 2017*).

For time-lapse live cell imaging, mouse ES cells expressing GFP-SLD5 from the endogenous locus were grown on 'µ-Dish 35 mm, high' (Ibidi, 81156) with 'no phenol red DMEM medium' (Thermo Fisher Scientific, 21063029) supplemented as described above. Cells were treated as above with LRR1 siRNA for a total of 48 hr, followed by MLN-4924 for 5 hr, to inhibit CUL2^LRR1 so that CMG would persist on chromatin until mitosis. Datasets were then acquired every 6 min for 14 hr (2 × 2 binning, 26% power of 0.26-s exposure at 488 nm laser to excite GFP) at approximately 30 different multiple stage positions, with heating to 37°C and 5% $CO_2$.

Subsequently, cells entering mitosis were identified by manual examination of time-lapse movie data. The time of nuclear envelope breakdown was indicated by dispersion of the nuclear GFP-SLD5 signal into the cytoplasmic area as shown in *Figure 6C* and observed previously (*Villa et al., 2021*). The persistence of GFP-SLD5 foci after nuclear envelope breakdown was monitored in each time-lapse series, until the time point at which the foci disappeared (signal no longer visible above the background fluorescence of the mitotic cytoplasm).

Only one cell line could be recorded per day, with each recording session typically yielding 50–150 cells that entered mitosis during the experiment. remained in focus throughout and could therefore be quantified subsequently. For each of the genotypes shown in *Figure 6D* and *Figure 6—figure supplement 4E*, three independent experiments were performed. Mean values were then calculated for each genotype, together with the average of the mean values and the associated standard deviation, as indicated in *Figure 6D* and *Figure 6—figure supplement 4E*. The samples were then compared via a Kruskal–Wallis test followed by Dunn's test, in order to assess statistical significance, using Prism 9 software (GraphPad, https://www.graphpad.com/scientific-software/prism/).

### Cell survival assay for mouse ES cells

Five hundred cells (except 1000 cells for FAF1Δ UBXN7Δ in *Figure 7A, B*) were plated into 10-cm Petri dishes. On the next day, 0, 3, or 6 μl of 10 mM MLN-4924 or DMSO (dimethyl sulfoxide) were added to 10 ml medium to give a final concentration of 0, 3, or 6 μM, before incubation for 6 hr. The drug-containing medium was then replaced with fresh medium, and incubation proceeded for a further 6 days. Cells were then washed with PBS and fixed with methanol. The fixed cells were stained with 0.5% crystal violet solution (Sigma-Aldrich, HT90132), and images of the plates were captured with a scanner. The number of colonies was counted manually, and the survival rate was then calculated.

### Statistics and reproducibility

No statistical method was employed to predetermine sample size and investigators were not blinded to allocation during the performance of experiments and assessment of results.

*Figure 6D*, *Figure 6—figure supplement 3F*, *Figure 6—figure supplement 4E*, and *Figure 7B–D* were repeated three times. All other experiments were repeated at least twice, except for *Figure 3—figure supplement 2C*, *Figure 4—figure supplement 1*, *Figure 5—figure supplement 1*, *Figure 6—figure supplement 3D, E*, and *Figure 6—figure supplement 4C*, which were performed once.

Graphs were generated and statistical tests performed using Prism 9 software (GraphPad, https://www.graphpad.com/scientific-software/prism/), as described above and in the figure legends. A Kruskal–Wallis test followed by Dunn's test was used to analyse the data in *Figure 6D* and *Figure 6—figure supplement 4E*, which represent multiple non-parametric samples. A two-tailed $t$-test was used in *Figure 7*, to assess statistical significance between two groups of data.

## Acknowledgements

We are grateful for the support of the Medical Research Council (core grant MC_UU_12016/13) and Cancer Research UK (Programme Grant C578/A24558 to KL and PhD studentship C578/A25671 to CPR). RF is supported by a JSPS Overseas Research Fellowship from the Japan Society for the Promotion of Science (202160572). We thank Tom Deegan for help with reconstituted ubiquitylation reactions with budding yeast proteins, Yisui Xia for help with equivalent reactions involving *C. elegans* factors, Lee Armstrong and Yogesh Kulathu for recombinant K48-linked ubiquitin chains and Fabrizio Villa for assistance with mouse ES cell work. We are also grateful to Axel Knebel and Clare Johnson for purified Uba1 and Ubiquitin and MRC PPU Reagents and Services (https://mrcppureagents.dundee.ac.uk) for antibody production.

## Additional information

### Funding

| Funder | Grant reference number | Author |
| --- | --- | --- |
| Medical Research Council | MC_UU_12016/13 | Karim PM Labib |
| Cancer Research UK | C578/A24558 | Karim PM Labib Ryo Fujisawa |
| Japan Society for the Promotion of Science | 202160572 | Ryo Fujisawa |
| Cancer Research UK | C578/A25671 | Karim PM Labib Cristian Polo Rivera |

The funders had no role in study design, data collection, and interpretation, or the decision to submit the work for publication.

### Author contributions

Ryo Fujisawa, Conceptualization, Formal analysis, Validation, Investigation, Visualization, Methodology, Writing – review and editing; Cristian Polo Rivera, Investigation; Karim PM Labib, Conceptualization, Supervision, Funding acquisition, Visualization, Methodology, Writing – original draft, Project administration

**Author ORCIDs**
Ryo Fujisawa http://orcid.org/0000-0003-1985-1668
Cristian Polo Rivera http://orcid.org/0000-0002-1433-4111
Karim PM Labib http://orcid.org/0000-0001-8861-379X

**Decision letter and Author response**
Decision letter https://doi.org/10.7554/eLife.76763.sa1
Author response https://doi.org/10.7554/eLife.76763.sa2

## Additional files

**Supplementary files**
• Transparent reporting form
• Supplementary file 1. Reagents and resources used in this study.
• Supplementary file 2. Plasmids generated in this study.

**Data availability**

All data generated or analysed during this study are included in the manuscript and supporting files; Source Data files have been provided .

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
