## [Editor Report]

The p97/Cdc48 AAA ATPase and its heterodimeric ubiquitin adapter Ufd1-Npl4 unfold ubiquitylated proteins, often to segregate large protein complexes such as the MCM helicase at the termination of replication. The authors now demonstrate that an important difference exists between the yeast and metazoan system. While Cdc48-Ufd1-Npl4 can target MCM7 with a relatively short ubiquitin chain (5-7 units), the metazoan p97 requires either much longer chains or one of a group of accessory proteins that have previously been connected to various p97 and Ufd1-Npl4 mediated processes. The paper presents multiple lines of experimental evidence in support of the proposed ubiquitin-chain length model.

---

## [Decision Letter]

**Decision letter after peer review:**

Thank you for submitting your article "Multiple UBX proteins reduce the ubiquitin threshold of the mammalian p97-UFD1-NPL4 unfoldase" for consideration by *eLife*. Your article has been reviewed by 3 peer reviewers, one of whom is a member of our Board of Reviewing Editors, and the evaluation has been overseen by David Ron as the Senior Editor. The reviewers have opted to remain anonymous.

Summary:

The p97/Cdc48 AAA ATPase and its heterodimeric ubiquitin adapter Ufd1-Npl4 unfolds ubiquitylated proteins, often to segregate large protein complexes such as the MCM helicase at termination of replication. The authors now demonstrate that an important difference exists between the yeast and metazoan system. While Cdc48-Ufd1-Npl4 can target MCM7 with a relatively short ubiquitin chain (5-7 units), the metazoan p97 requires either much longer chains or one of a group of accessory proteins that have previously been connected to various p97 and Ufd1-Npl4 mediated processes.

Essential revision:

The reviewers agreed that the paper has several strengths, including the very nice in vitro system for analysis of the role of p97 in segregase activity on a multi-subunit complex. Some aspects of the biochemistry are strong, and the system itself – CMG disassembly – is embedded in rich biology, which is also a plus. However, the reviewer's feel that the paper falls short of fully demonstrating the proposed mechanism, and several weaknesses were noted (as provided in the detailed reviewer comments). In particular, given where the p97 field is currently, further information is needed to fully understand the actual biochemical mechanism of UBX-dependent enhancement of unfolding. Thus, we would ask for major revisions.

After discussion among the reviewers, the consensus is that for further consideration, the following points would need to be addressed. We realize that this represents a substantial revision to the paper, with significant experimental aspects needing to be addressed.

1. The description of the mechanism by which the UBX domain (together with the coiled-coil) can promote unfolding needs further elaboration. Further experiments could employ crosslink sensors of protein unfolding coupled with binding assays to understand where/how in the unfolding process the UBX effect is manifested. Without this, the reader is left wondering what is actually happening biochemically. Given the advanced stage of our understanding of p97 Ub recognition and unfolding, some additional mechanistic understanding is relevant in this case.

2. The reviewer's also commented on the treatment of the Ub chains in your analysis. Given that the primary conclusion is that Ub chain length is relevant, a more definitive description of the chains on the different forms of the CMG being examined. There are various aspects here, ranging from length to chain topology. Given that you have used Cdc34 to build chains on CMG, this is almost certainly exclusively K48-linked chains, so perhaps the topology aspect isn't as import as other aspects. Related to this is the question of whether the differential effects seen on unfolding with apparent chain length reflects chain length, weak binding, or a differential intrinsic activity of yeast versus human p97. One suggestion is to perform kinetic experiments in order to address this critical question.

3. There are significant concerns about the viability and survival assays in Figure 6. For the cell-based SLD5 mobilization assay, you would need to: (1) discuss how they quantified (preferably automated), (2) depict the data in a box blot for that type of measurement, and (3) show significance with an appropriate test. More importantly, the data as it stands only shows the requirement of the accessory protein, which has been shown in several other papers previously. Ultimately, to bring the story full circle, you would need to demonstrate that the structural components of the UBX protein (UBX domain and coiled-coil) do (or do not) rescue the various cellular phenotypes. Given that the actual function of the coiled coil and the UBX haven't been mapped to specific residues, then it seems that some understanding of these elements would be needed.

Reviewer #1 (Recommendations for the authors):

This paper describes a systematic biochemical analysis of UBX proteins in facilitating protein unfolding by the p97-UFD1-NPL4 (referred to here is the p97 complex). The p97 complex binds Ub and unfolds it to allow the ubiquitylated protein to be translocated into the p97 ATPase pore for unfolding. This paper demonstrates that UBX proteins are able to reduce the necessary ubiquitin chain length in order to support unfolding by p97. They explore this using ubiquitylated CMG helicase as a substrate. Removal of CMG helicase from replicated DNA is required for completion of DNA synthesis.

First the authors demonstrate that the p97 complex only only unfolds CMG with very long Ub chains. The then show that the high threshold for Ub is reduced when UBXN7, FAF1 or FAF2 are added. These proteins bind to both the p97 complex and Ub in substrates. This is then followed up in cells by demonstrating that removal of UBXN7 and FAF1 reduces CMG disassembly and is synthetic with reduced CMG ubiquitin ligase activity.

1. I found the section describing Figure 2 to be very unclear (especially lines 152-155). It sort of seems like 2 things are being changed at once and interpreted in a way that isn't very clearly described. First, the interpretation of Low versus High conditions and extent of ubiquitylation needs to be justified better. It appears to be based on prior studies but has the site(s) of ubiquitylation in the "Low" sample been demonstrated independently for the samples used here? This seems critical to validate, since if for some reason, the Low sample has substantial multi-mono versus mostly ubiquitylation on a single site (K29), the interpretation would be different.

The description of the results is also sufficiently terse as to not be clear as to what is really being measured and concluded. It is conceivable that the processivity of the CRL in this case (i.e. the actual chain length) varies with the primary ubiquitylation site. Thus, it may not be easy to compare the apparent efficiency of different sites as the length of chains on each site may be different. It would be nice if there was a way to chain amputate the UB chains and measure the average length of chains for the 2 populations of substrates. Proteasomes from yeast lacking Ubp6 should release Ub chains from the Mcm7 ubiquitylated species as "amputated" chains via Rpn11 activity, and so in the in vitro setting, it might be possible to determine the constellation of chain length using anti-Ub to blot reactions. An alternative would be to use Ub-AQUA proteomics or possibly the Clippase enzyme to measure the average chain length for each population. If there are several sites of ubiquitylation in WT MCM7, the chains themselves might not be significantly longer than on the mostly K29-modified sample, but the authors seem to want to conclude its chain length that is the relevant thing changing.

In terms of the interpretation of the results – it seems pretty clear that for the K29 only/Low sample, Cdc48 prefers the longer forms but for the WT/High sample, it looks like Cdc48 does not prefer the longest forms, as there is a slight bias towards longer forms for the protein left on the beads (if reproducible).

An additional issue for this analysis concerns whether the intrinsic or specific activity of p97 and Cdc48 for ATP binding, hydrolysis and unfolding are equivalent. This is especially important for interpreting the activity of p97 towards the "Low" sample. Do the authors have a way to measure the specific activity of their p97 and Cdc48 preps? The proteins themselves could be differentially prone to damage or inactivation during purification, thereby creating distinct specific activities for the two preps that could affect the interpretation.

In the end, I am not sure that the conclusion "These findings indicated that human p97-UFD1-NPL4 has a much higher ubiquitin threshold than yeast Cdc48-Ufd1-Npl4." Is that strong based on the data in Figure 2.

Also, in figure 2A and text, I am not sure the words "limited" and "efficient" are the best terms to use. Chain lengths may be comparable in both cases. Maybe something like "site-specific" for the K29-only version and "multi-site" for the other.

Given the inability of the p97 complex to unfold the CMG under "Low" ubiquitylation conditions, it might be worth asking whether increasing the amount of p97 complex in the reaction can overcome the barrier. This could potentially address issues related to reduced specific activity. It could also potentially address reduced Ub affinity for the p97 complex versus the Cdc48 complex.

2. The data in Figure 3 are said to be based on CMG complexes that had up to ~12 ubiquitin conjugated to the Mcm7 subunit. This figure is also subject to the same issue as above – can the authors distinguish between one long chain versus some amount of multi-site ubiquitylation and shorter chains in this case?

3. The deletion analysis in Figure 4 doesn't seem to example the extent to which the various mutants assemble with the p97 complex. One presumes a correlation, but this wasn't tested directly.

One line 216, it is stated: "These findings indicated that the coiled-coil domain of FAF1 has a previously unanticipated role in stimulating the unfoldase activity of p97-UFD1-NPL4." However, it is possible that without this coiled-coil, the UBX domain doesn't associate effectively with the p97 complex. This would be sort of a trivial result, in that there isn't a "direct" role of the copied-coil, but it is simply required for the UBX domain to interact. Can the coiled-coil be replaced by an alternative coiled-coil sequence or is there something special about this coiled-coil?

4. Although a bit hard to know for sure, it looks like the ubiquitylated forms of Mcm7 in the ES cell experiment (Figure 5) have much shorter chains than the experiments with yeast CMG – especially in 5b. Of course it is possible that this reflects the presence of DUB activity during the IP, which would tend to shorten the chains. But at face value, it looks like FAF1/UBXN7 can allow p97 to operate on very short chains (although one really cant distinguish this from multi-mono ubiquitylation).

This raises questions about the overall conclusion. If p97 inhibitor increases the appearance of MCM7 conjugates (albeit with shorter chains) in the FAF1/UBXN7 knockout substantially, this suggest that p97 can disassemble shorter chains in cells without an adaptor OR that there is an additional UBX protein in cells that can do the job. Of course if the ubiquitylation levels are a result of DUB activity, that could explain it but it seems like this should be ruled out.

*Reviewer #2 (Recommendations for the authors):*

1) The authors nail down the critical activity to the coiled-coil domain of the accessory factors FAF1/FAF2 (in addition to the UBX domain that binds p97). This is surprising and interesting because the UBA domains of FAF1 or FAF2 are not required. Frustratingly, it is not revealed what the function of the coiled-coil region is, which limits the mechanistic understanding. The authors speculate that the coiled coil region binds ubiquitin. This needs to be tested because it is a simple experiment. The comparison to Pup is inappropriate because Pup is not homologous to ubiquitin. Alternatively, the coiled coil region could modulate p97 or Ufd1-Npl4 binding activity. The question would be whether the coiled coil region binds p97 and/or Ufd1-Npl4. Technically, this can be tested.

2) The biochemical experiments show single time points. In contrast, the field has reached a point where kinetic data can be provided. That would allow to quantitatively define even smaller differences such as the deltaUBA or UIM effect in Figure 4C

3) The ES cell microscopy in Figure 5C needs to be quantified by an automated, unbiased pipeline.

4) Figure 6, cell viability and survival: The information gleaned from these experiments is questionable. Effects on colony formation are mostly mediated by MLN4924. The additional effect by the deletions is minor, and needs to be quantified and statistically validated, also to clarify whether it is additive or synthetic. The same applies to the viability assay. More importantly, it is impossible to connect the effects to replication termination. Firstly, there are many processes affected by the manipulations. Secondly, the MLN4924 treatment is so short that it will affect replication termination only in a small percentage of cells.

[Editors' note: further revisions were suggested prior to acceptance, as described below.]

Thank you for resubmitting your work entitled "Multiple UBX proteins reduce the ubiquitin threshold of the mammalian p97-UFD1-NPL4 unfoldase" for further consideration by *eLife*. Your revised article has been evaluated by David Ron (Senior Editor) and a Reviewing Editor.

The manuscript has been improved but there are some remaining issues that need to be addressed, as outlined below:

There are just a couple of small items that need attention – noting the p-value for Figure 6D is weakly significant and then a couple of items mentioned by reviewer 2. The paper will then be ready to accept.

*Reviewer #1 (Recommendations for the authors):*

Overall, the authors have done extensive work to try to address the previous reviewers' comments. The authors have addressed all of the specific comments, and in general, the results support the conclusions. One issue is that in Figure 6D, the p-value is only 0.09, which is not strongly significant. This may have to do with the assay itself. The majority of experiments employ ubiquitin smears to monitor p97 activity as it relates to the length dependence on the biological processes.

*Reviewer #2 (Recommendations for the authors):*

The authors have addressed concerns regarding data quality and evaluation. Moreover, they added a lot of valuable data on how the Ufd1-Npl4 adapter determines the choice of Ub chain length, including the Npl4 groove mutations and the very neat and informative "species swap". The dissection of effects of Ub chain number, branching, and type is very nice. They also added some more informative biochemical characterization of the critical coiled-coil region of FAF1, and I agree with the authors that more detailed mechanistic dissection goes beyond this analysis. This is a very rich and careful study on a timely and important question in replication termination and p97 function.

*Reviewer #3 (Recommendations for the authors):*

The authors performed a large number of additional experiments adequately addressing all key points of the Essential Revision section of the decision letter. Moreover, they addressed all specific criticisms raised by this reviewer satisfactorily. The revised manuscript is substantially improved and contains important new data on mechanistic aspects of the observed 'ubiquitin threshold'. Acceptance for publication is strongly recommended.

---

## [Author Response]

Essential revision:The reviewers agreed that the paper has several strengths, including the very nice in vitro system for analysis of the role of p97 in segregase activity on a multi-subunit complex. Some aspects of the biochemistry are strong, and the system itself – CMG disassembly – is embedded in rich biology, which is also a plus. However, the reviewer's feel that the paper falls short of fully demonstrating the proposed mechanism, and several weaknesses were noted (as provided in the detailed reviewer comments). In particular, given where the p97 field is currently, further information is needed to fully understand the actual biochemical mechanism of UBX-dependent enhancement of unfolding. Thus, we would ask for major revisions.After discussion among the reviewers, the consensus is that for further consideration, the following points would need to be addressed. We realize that this represents a substantial revision to the paper, with significant experimental aspects needing to be addressed.1. The description of the mechanism by which the UBX domain (together with the coiled-coil) can promote unfolding needs further elaboration. Further experiments could employ crosslink sensors of protein unfolding coupled with binding assays to understand where/how in the unfolding process the UBX effect is manifested. Without this, the reader is left wondering what is actually happening biochemically. Given the advanced stage of our understanding of p97 Ub recognition and unfolding, some additional mechanistic understanding is relevant in this case.

To elaborate further the action of the UBX proteins, we first needed to clarify the mechanistic basis for the difference in ubiquitin threshold between human p97-UFD1-NPL4 and yeast Cdc48-Ufd1-Npl4. Taking advantage of the very high sequence conservation between human p97 and yeast Cdc48 (Figure 3—figure supplement 1A of the revised manuscript), we performed reconstituted unfoldase assays that tested the ability of human p97 to work with yeast Ufd1-Npl4, and yeast Cdc48 to function with human UFD1-NPL4. In this way, we tested whether the difference in ubiquitin threshold between yeast Cdc48-Ufd1-Npl4 (five or more ubiquitins) and human p97-UFD1-NPL4 (many ubiquitins) was due to differences between Sc Cdc48 & Hs p97, Sc Ufd1-Npl4 & Hs UFD1-NPL4, or both. Despite the evolutionary distance between humans and yeast, such ‘species swap’ experiments worked well and the new data in Figure 3A show that all combinations of Sc Cdc48 / Hs p97 with Sc Ufd1-Npl4 / Hs UFD1-NPL4 supported efficient disassembly of an extensively ubiquitylated CMG substrate. Crucially, however, human UFD1-NPL4 did not support the disassembly of CMG with 5-15 ubiquitins conjugated to the CMG-Mcm7 subunit, either in reactions with human p97 or yeast Cdc48 (Figure 3B, lanes 5-8 and Figure 4—figure supplement 3, lanes 3-4 and 7-8). In contrast, yeast Ufd1-Np4 allowed both yeast Cdc48 (Figure 3B lanes 3-4) and human p97 (Figure 3B lanes 9-10) to disassemble CMG with five or more ubiquitins conjugated to CMG-Mcm7. These data demonstrate that the different ubiquitin thresholds of human p97-UFD1-NPL4 and yeast Cdc48-Ufd1-Npl4 are determined by differences between human UFD1-NPL4 and yeast Ufd1-Npl4, rather than reflecting differences in the enzymatic activity of human p97 and yeast Cdc48.

Human UFD1-NPL4 and yeast Ufd1-Npl4 are less well conserved in primary sequence than human p97 and yeast Cdc48 (Figure 3—figure supplement 1A-C). Nevertheless, previous structural studies showed that human UFD1-NPL4 and yeast Ufd1-Npl4 interact in an analogous manner with human p97 and yeast Cdc48 (Bodnar et al., 2018; Pan et al., 2021). Moreover, both yeast Ufd1-Npl4 and human UFD1-NPL4 share conserved interaction sites with K48-linked polyubiquitin, via the top of the Sc Npl4 / Hs NPL4 ‘tower’ that sits on top of the N-terminal ring of p97 adjacent to the central pore, and the UT3-domain of Sc Ufd1 / Hs UFD1 (discussed on lines 220-227 of the revised manuscript). Elegant studies of yeast Cdc48-Ufd1-Npl4 from Tom Rapoport’s group showed that binding of yeast Ufd1-Npl4 to K48-linked polyubiquitin leads to capture of an unfolded ubiquitin intermediate via a groove on the outside of the Npl4 tower, which thereby directs unfolded ubiquitin into the Cdc48 central pore (Twomey et al., 2019). Correspondingly, mutations of conserved residues in the groove of yeast Npl4 block substrate unfolding (Twomey et al., 2019). We now show in the revised manuscript that the equivalent mutations in human NPL4 (Figure 3—figure supplement 2A-B, NPL4-AAE) block the disassembly of ubiquitylated CMG (Figure 3—figure supplement 2C lanes 7-8), without affecting the binding of NPL4 to UFD1 (Figure 3—figure supplement 2B) or p97 (Figure 3—figure supplement 2D). Together with the previous studies mentioned above and discussed in our manuscript, this indicates that human p97-UFD1-NPL4 also initiates substrate unfolding via an unfolded ubiquitin intermediate, just like yeast Cdc48-Ufd1-Npl4.

However, previous studies also point to functional diversification between human UFD1-NPL4 and yeast Ufd1-Npl4. In particular, human NPL4 has a ubiquitin-binding Zn finger at its C-terminus (NPL4-NZF) that is not found in yeast and represents the highest affinity binding site for ubiquitin in human UFD1-NPL4. Nevertheless, the NPL-NZF was previously shown to be dispensable in human cells for the ER-associated degradation of misfolded and ubiquitylated proteins (Ye et al., 2003). This suggested that the evolutionarily conserved low-affinity binding sites for K48-linked ubiquitin chains in UFD1-NPL4, at the top of the NPL4 tower and in the UT3 domain of UFD1 (Ji et al., 2021; Pan et al., 2021) (Park et al., 2005; Sato et al., 2019; Twomey et al., 2019; Ye et al., 2003), are sufficient to initiate the unfolding of ubiquitylated substrates of p97. Consistent with past work (Meyer et al., 2002), we show in the revised manuscript that deletion of the NPL4-NZF did not impair association with p97 or UFD1 (Figure 3—figure supplement 2B and D) but abrogated the detectable interaction of UFD1-NPL4 with K48-linked ubiquitin chains in vitro (Figure 3—figure supplement 2E). Importantly, we show that human p97-UFD1-NPL4-∆NZF still supports the disassembly of heavily ubiquitylated CMG helicase in reconstituted in vitro assays, albeit with reduced efficiency compared to wild type NPL4 (Figure 3—figure supplement 2C, compare lanes 1-6). Considered together with the data and previous observations discussed above, these findings indicate that the dynamic association of UFD1-NPL4 with very long K48-linked ubiquitin chains, likely via the conserved low affinity ubiquitin-binding sites in the NPL4 tower and UFD1-UT3 domain, is sufficient to initiate substrate unfolding, dependent upon the NPL4 groove. Additional ubiquitin binding by the NPL4 NZF domain is not an essential part of the mechanism of substrate unfolding, but likely increases the efficiency of substrate engagement by human p97-UFD1-NPL4, as does increased length of the K48-linked ubiquitin chains (see below for a detailed characterisation of ubiquitin chains in the revised manuscript).

The above insights into the mechanism of human p97-UFD1-NPL4, and the role of UFD1-NPL4 in setting the ubiquitin threshold, provide a framework for understanding how UBX proteins stimulate the ability of human p97-UFD1-NPL4 to process substrates with shorter K48-linked ubiquitin chains. CMG disassembly by human p97-UFD1-NPL4 in the presence of UBX proteins required five or more ubiquitins on the CMG-Mcm7 subunit (Figure 4E lanes 7-8), thereby matching the minimal ubiquitin threshold of yeast Cdc48-Ufd1-Npl4 (Figure 4E lanes 3-4) that is thought to reflect the presence of multiple conserved ubiquitin-binding domains within yeast Ufd1-Npl4 (Deegan et al., 2020; Park et al., 2005; Sato et al., 2019; Twomey et al., 2019). This suggested that the UBX proteins function by stimulating productive interactions between the multiple ubiquitin-binding modules of human p97-UFD1-NPL4 and ubiquitylated substrate proteins. The initial engagement of UFD1-NPL4 with ubiquitylated substrates in the presence of UBX proteins should then lead to the trapping of an unfolded ubiquitin intermediate on the NPL4 groove. Consistent with this view, we now show in the revised manuscript that CMG disassembly in the presentence of UBXN7, FAF1 and FAF2 is dependent upon the NPL4 groove (Figure 4—figure supplement 2B; compare lanes 5-6 and 13-14).

We also show in the revised manuscript that UBX proteins suppressed the partial defect in disassembling heavily ubiquitylated CMG that was observed in the absence of the ubiquitin-binding Zinc finger at the carboxyl terminus of NPL4 (Figure 4—figure supplement 2, compare release of Mcm6 and Sld5 into supernatant in lanes 5-10). Moreover, UBX proteins restored the ability of yeast Cdc48 to disassemble CMG complexes in the presence of human UFD1-NPL4, when the K48-linked ubiquitin chains on CMG-Mcm7 were otherwise too short to allow disassembly (Figure 4—figure supplement 3, lanes 1-6). These findings indicated that the UBX proteins augment the ability of human UFD1-NPL4 to initiate the unfolding of ubiquitylated substrates by p97.

The UBXN7 protein is predominantly nuclear in mammalian cells and our manuscript showed that UBXN7 stimulates the disassembly of ubiquitylated CMG by p97-UFD1-NPL4, both in reconstituted assays with purified proteins (Figure 4D) and in mouse ES cells (Figure 6; Figure 6—figure supplement 4). The reconstituted assays indicate that the ability of UBXN7 to stimulate p97-UFD1-NPL4 requires both the UBX domain that binds p97, and the UBA domain that binds ubiquitin (Figure 5A-C; Figure 5—figure supplement 1A-B show that the UBXN7-∆UBA still interacts with p97-UFD1-NPL4). Together with the data discussed above, these findings indicate that UBXN7 functions by binding to p97-UFD1-NPL4 and stabilising productive interactions between UFD1-NPL4 and K48-linked ubiquitin chains, which then leads firstly to trapping of an unfolded ubiquitin intermediate on the NPL4 groove and then to translocation of the substrate polypeptide through the p97 central channel.

The situation with FAF1 (and FAF2) is more complex, since stimulation of p97-UFD1-NPL4 is dependent upon a coiled coil domain adjacent to the UBX domain, in addition to the UBX domain itself. We show in the revised manuscript that the FAF1 coiled coil is not required for association of the FAF1-UBX with p97-UFD1-NPL4 (Figure 5—figure supplement 1E-F). Nor does the coiled coil of FAF1 bind detectably to UFD1-NPL4 in the absence of p97 (Figure 5—figure supplement 1E-F). Furthermore, human p97 associates with UFD1-NPL4 in the absence of UBX proteins (Figure 3—figure supplement 2D), as shown previously (Meyer et al., EMBO J, 2002). We could not detect stable binding of the FAF1 coiled-coiled to K48-linked ubiquitin chains in vitro (Figure 5—figure supplement 3, FAF1-∆268 and FAF1-∆480). However, the same is true for UFD1-NPL4-∆NZF (Figure 3—figure supplement 2), which nevertheless contains conserved low affinity ubiquitin-binding modules, which we show are sufficient to support the disassembly of ubiquitylated CMG in reconstituted assays (Figure 3—figure supplement 2C). Therefore, we cannot exclude that the FAF1 coiled coil also functions by binding to both p97-UFD1-NPL4 and ubiquitin, with ubiquitin binding being low affinity yet functionally relevant. In our opinion, further mechanistic insight into the action of the coiled coil domain will require structural studies of early intermediates in the unfolding reaction, which are considerably beyond the scope of the present study.

In summary the manuscript provides the following new mechanistic insights for mammalian p97-UFD1-NPL4:

– Substrate unfolding by human p97-UFD1-NPL4 is dependent on the NPL4 groove and thus is likely to initiate via an unfolded ubiquitin intermediate, as for yeast Cdc48-Ufd1-Npl4.

– Human UFD1-NPL4 sets a high ubiquitin threshold for substrate processing by human p97, likely reflecting dynamic association of the conserved ubiquitin-binding modules in the NPL4 tower and UFD1 UT3-domain with K48-linked ubiquitin chains.

– A set of human UBX proteins reduce the ubiquitin threshold of human p97-UFD1-NPL4 to the same level as yeast Cdc48-Ufd1-Npl4. The same UBX proteins also stimulate yeast Cdc48 in combination with human UFD1-NPL4. Stimulation by the UBX proteins is dependent on the NPL4 groove that in yeast is known to bind unfolded ubiquitin during initiation of unfolding. Overall, these findings indicate that the human UBX proteins stabilise productive interactions between the multiple conserved ubiquitin binding sites in Hs UFD1-NPL4 and K48-linked polyubiquitin, thereby promoting the initiation of substrate unfolding via an unfolded ubiquitin intermediate.

– Consistent with the above model, stimulation of p97-UFD1-NPL4 by UBXN7 requires both the p97-binding UBX domain and the ubiquitin-binding UBA domain. These findings provide proof of principle that a UBX protein can stimulate p97-UFD1-NPL4, by binding to the unfoldase and supplying additional ubiquitin binding, which then facilitates the initiation of ubiquitin unfolding by UFD1-NPL4, dependent on the NPL4 groove.

2. The reviewer's also commented on the treatment of the Ub chains in your analysis. Given that the primary conclusion is that Ub chain length is relevant, a more definitive description of the chains on the different forms of the CMG being examined. There are various aspects here, ranging from length to chain topology. Given that you have used Cdc34 to build chains on CMG, this is almost certainly exclusively K48-linked chains, so perhaps the topology aspect isn't as import as other aspects.

The revised manuscript contains new data that demonstrate clearly how CMG-Mcm7 is ubiquitylated under the various conditions employed in our study:

– Figure 1—figure supplement 1A indicates that 2-3 ubiquitin chains are conjugated to the CMG-Mcm7 subunit in the presence of high concentrations of E2 (25 nM) and E3 (10 nM), since 2-3 ubiquitins are conjugated to CMG-Mcm7 in reactions performed with lysine-free ubiquitin (Figure 1—figure supplement 1A, K0 ubiquitin). Under these very efficient conditions, much of the chain-linkage is via K48 of ubiquitin but other linkages are also involved (Figure 1—figure supplement 1A, reactions with K48R ubiquitin).

– Figure 1—figure supplement 1B indicates that a single ubiquitin chain of up to ~15 ubiquitins is conjugated to the CMG-Mcm7 subunit in the presence of low concentrations of E2 (2.5 nM) and E3 (1 nM), since a single ubiquitin is conjugated to CMG-Mcm7 in reactions performed with lysine-free ubiquitin (Figure 1—figure supplement 1B, K0 ubiquitin). Under these less efficient conditions, essentially all of the chains are linked via K48 of ubiquitin (Figure 1—figure supplement 1B, compare reactions with K0 ubiquitin and K48R ubiquitin). Doubling the concentration of E3 under such conditions produces some longer chains on CMG-Mcm7 (Figure 1—figure supplement 1D, 2.5 nM E2 and 2 nM E3) and the linkages are still almost entirely via K48 of ubiquitin (Figure 1—figure supplement 1D, compare reactions with K0 ubiquitin and K48R ubiquitin).

Human p97-UFD1-NPL4 disassembles extensively ubiquitylated CMG (25 nM E2 and 10 nM E3). In the revised manuscript we show that the action of p97-UFD1-NPL4 under such conditions is not dependent upon the presence of 2-3 ubiquitin chains on CMG-Mcm7, using a new allele with lysine mutations in Mcm7 that limit ubiquitylation to a single chain on Mcm7 (Figure 1—figure supplement 1C, CMG-Mcm7-KR, compare wt ubiquitin and K0 ubiquitin). The very long chain on CMG-Mcm7 under such conditions includes much K48-linkage of ubiquitin but with other linkages too (Figure 1—figure supplement 1C, compare wt ubiquitin and K48R ubiquitin). Disassembly of CMG-Mcm7-KR with a single long ubiquitin chain is equally efficient as when 2-3 ubiquitin chains are conjugated to wild type CMG-Mcm7 (Figure 2—figure supplement 3A). Moreover, human p97-UFD1-NPL4 and yeast Cdc48-Ufd1-Npl4 show a similar ability to disassemble CMG complexes ubiquitylated with ‘K48-only ubiquitin’ that can only form K48-linked chains, either under conditions with 2-3 chains per CMG-Mcm7 subunit (Figure 2—figure supplement 3B, lanes 1-6, wild type Mcm7) or under conditions with just one long K48-linked chain conjugated to CMG-Mcm7-KR (Figure 2—figure supplement 3B, lanes 7-12, CMG-Mcm7-KR).

Overall, the above data indicate that human p97-UFD1-NPL4 preferentially disassembles CMG complexes that have long K48-linked ubiquitin chains conjugated to CMG-Mcm7, without any requirement for multiple chains or other chain linkages.

Related to this is the question of whether the differential effects seen on unfolding with apparent chain length reflects chain length, weak binding, or a differential intrinsic activity of yeast versus human p97.

Please see the reply to ‘essential point 1’ above, where we discuss our new ‘species swap’ experiments that show that the high ubiquitin threshold of human p97-UFD1-NPL4 is due to the properties of UFD1-NPL4, rather than due to differential intrinsic activities of yeast Cdc48 versus human p97 (Figure 3).

We also show that deletion of the ubiquitin-binding Zn finger of NPL4 has only a mild effect on CMG disassembly (Figure 3—figure supplement 2A-C), though in its absence the interaction of UFD1-NPL4 with K48-linked ubiquitin chains is highly dynamic (Figure 3—figure supplement 2E). These findings indicate that the high ubiquitin threshold of human p97-UFD1-NPL4 reflects the dynamic interaction of the conserved ubiquitin-binding modules of UFD1-NPL4 (within the NPL4 tower and the UFD1 UT3 domain) with K48-linked ubiquitin, with long chains compensating for such dynamic interactions.

One suggestion is to perform kinetic experiments in order to address this critical question.

The revised manuscript presents a new kinetic experiment in Figure 2—figure supplement 1. Under conditions where the majority of extensively ubiquitylated CMG complexes are disassembled within ~10 minutes, very few CMG complexes with up to ~10 ubiquitins are disassembled even by 30 minutes.

Moreover, Figure 2—figure supplement 2 presents a titration of p97-UFD1-NPL4. Whereas the majority of extensively ubiquitylated CMG complexes were disassembled in the presence of 15 nM p97-UFD1-NPL4, very little disassembly of CMG complexes with up to ~12 ubiquitins is observed even in the presence of 150 nM p97-UFD1-NPL4.

3. There are significant concerns about the viability and survival assays in Figure 6. For the cell-based SLD5 mobilization assay, you would need to: (1) discuss how they quantified (preferably automated), (2) depict the data in a box blot for that type of measurement, and (3) show significance with an appropriate test. More importantly, the data as it stands only shows the requirement of the accessory protein, which has been shown in several other papers previously. Ultimately, to bring the story full circle, you would need to demonstrate that the structural components of the UBX protein (UBX domain and coiled-coil) do (or do not) rescue the various cellular phenotypes. Given that the actual function of the coiled coil and the UBX haven't been mapped to specific residues, then it seems that some understanding of these elements would be needed.

Full details of data quantification for microscopy experiments are now provided in Materials and methods (lines 1061-1079). We also include a new section of Materials and methods entitled Statistics and Reproducibility.

In the original version of the manuscript, Figure 5D contained a histogram that summarised the GFP-SLD5 data representing the mitotic pathway for CMG helicase disassembly. In the revised version of the manuscript, we have replaced the histogram with a scatter plot (Figure 6D in the revised version) that presents the raw data together with the mean values from three independent experiments, the average of the three means, and a statistical analysis of the data via a Kruskal-Wallis test followed by Dunn’s test. The associated p values are now shown in Figure 6D.

We also present new data in Figure 6—figure supplement 4, whereby the defects of *UBXN7∆ FAF1∆* cells are rescued by expression of full-length UBXN7, full length FAF1, FAF1-∆UBX or FAF1-∆CoiledCoil (FAF1-∆CC). By monitoring the accumulation of ubiquitylated CMG-MCM7 in UBXN7∆ FAF1∆ cells (Figure 6—figure supplement 4D), we show that wild type FAF1 rescues but FAF1-∆UBX and FAF1-∆CoiledCoil do not. Similarly, we monitored the mitotic CMG disassembly defect of *UBXN7∆ FAF1∆* cells by spinning disk confocal microscopy (Figure 6—figure supplement 4E) and showed that expression of wild type FAF1 produced a statistically significant rescue (as did expression of wild type UBXN7), but expression of FAF1-∆UBX or FAF1-∆CoiledCoil did not.

Reviewer #1 (Recommendations for the authors):This paper describes a systematic biochemical analysis of UBX proteins in facilitating protein unfolding by the p97-UFD1-NPL4 (referred to here is the p97 complex). The p97 complex binds Ub and unfolds it to allow the ubiquitylated protein to be translocated into the p97 ATPase pore for unfolding. This paper demonstrates that UBX proteins are able to reduce the necessary ubiquitin chain length in order to support unfolding by p97. They explore this using ubiquitylated CMG helicase as a substrate. Removal of CMG helicase from replicated DNA is required for completion of DNA synthesis.First the authors demonstrate that the p97 complex only only unfolds CMG with very long Ub chains. The then show that the high threshold for Ub is reduced when UBXN7, FAF1 or FAF2 are added. These proteins bind to both the p97 complex and Ub in substrates. This is then followed up in cells by demonstrating that removal of UBXN7 and FAF1 reduces CMG disassembly and is synthetic with reduced CMG ubiquitin ligase activity.1. I found the section describing Figure 2 to be very unclear (especially lines 152-155). It sort of seems like 2 things are being changed at once and interpreted in a way that isn't very clearly described. First, the interpretation of Low versus High conditions and extent of ubiquitylation needs to be justified better. It appears to be based on prior studies but has the site(s) of ubiquitylation in the "Low" sample been demonstrated independently for the samples used here? This seems critical to validate, since if for some reason, the Low sample has substantial multi-mono versus mostly ubiquitylation on a single site (K29), the interpretation would be different.

We take the reviewer’s point and in the revised manuscript we characterise carefully the ubiquitin chains on CMG-Mcm7 under the various conditions used (data discussed above in relation to ‘essential revision point 2’). The new data are presented in Figure 1—figure supplement 1, confirming that ‘low ubiquitylation conditions’ involve the conjugation on CMG-Mcm7 of a single K48-linked ubiquitin chain.

The description of the results is also sufficiently terse as to not be clear as to what is really being measured and concluded. It is conceivable that the processivity of the CRL in this case (i.e. the actual chain length) varies with the primary ubiquitylation site. Thus, it may not be easy to compare the apparent efficiency of different sites as the length of chains on each site may be different. It would be nice if there was a way to chain amputate the UB chains and measure the average length of chains for the 2 populations of substrates. Proteasomes from yeast lacking Ubp6 should release Ub chains from the Mcm7 ubiquitylated species as "amputated" chains via Rpn11 activity, and so in the in vitro setting, it might be possible to determine the constellation of chain length using anti-Ub to blot reactions. An alternative would be to use Ub-AQUA proteomics or possibly the Clippase enzyme to measure the average chain length for each population. If there are several sites of ubiquitylation in WT MCM7, the chains themselves might not be significantly longer than on the mostly K29-modified sample, but the authors seem to want to conclude its chain length that is the relevant thing changing.

See above – in the revised version we characterise carefully the ubiquitylation of CMG-Mcm7 under each condition, showing that ‘low ubiquitylation’ conditions (2.5 nM E2 and 1 nM E3) involve a single K48-linked chain on CMG-Mcm7, whilst ‘high ubiquitylation’ conditions (25 nM E2 and 10 nM E3) produce 2-3 chains per CMG-Mcm7, largely K48-linked but with other linkages too (Figure 1—figure supplement 1). Importantly, we show in Figure 2—figure supplement 3 that multiple chains on CMG-Mcm7 are not required for efficient disassembly by p97-UFD1-NPL4 (by mutating Mcm7 lysines so that only one ubiquitin chain is formed, even under highly efficient ubiquitylation conditions). We also show that yeast Cdc48-Ufd1-Npl4 and human p97-UFD1-NPL4 are both able to disassemble CMG that has been ubiquitylated with ‘K48 only ubiquitin’ (with other lysines mutated to arginine), indicating that the ability of human p97-UFD1-NPL4 to disassemble extensively ubiquitylated CMG is not due to a requirement for other chain linkages that are produced under such conditions (K48-linked chains are processed by both human and yeast unfoldases).

In terms of the interpretation of the results – it seems pretty clear that for the K29 only/Low sample, Cdc48 prefers the longer forms but for the WT/High sample, it looks like Cdc48 does not prefer the longest forms, as there is a slight bias towards longer forms for the protein left on the beads (if reproducible).

We take the reviewer’s point but think that this detail likely reflects the fact that extremely long ubiquitin chains tend to stick to beads non-specifically due to reduced solubility.

An additional issue for this analysis concerns whether the intrinsic or specific activity of p97 and Cdc48 for ATP binding, hydrolysis and unfolding are equivalent. This is especially important for interpreting the activity of p97 towards the "Low" sample. Do the authors have a way to measure the specific activity of their p97 and Cdc48 preps? The proteins themselves could be differentially prone to damage or inactivation during purification, thereby creating distinct specific activities for the two preps that could affect the interpretation.

In Figure 3 of the revised manuscript we perform ‘species swap experiments’ that indicate that the high ubiquitin threshold of human p97-UFD1-NPL4 reflects the properties of human UFD1-NPL4, rather than the properties of human p97. We show that yeast Cdc48 functions with human UFD1-NPL4 but the hybrid unfoldase now behaves like human p97-UFD1-NPL4. Similarly, yeast Ufd1-Npl4 works with human p97 and the hybrid unfoldase behaves more like yeast Cdc48-Ufd1-Npl4.

In the end, I am not sure that the conclusion "These findings indicated that human p97-UFD1-NPL4 has a much higher ubiquitin threshold than yeast Cdc48-Ufd1-Npl4." Is that strong based on the data in Figure 2.

We take the reviewer’s point but now provide extensive new data in the revised manuscript to establish the point, as discussed above.

Also, in figure 2A and text, I am not sure the words "limited" and "efficient" are the best terms to use. Chain lengths may be comparable in both cases. Maybe something like "site-specific" for the K29-only version and "multi-site" for the other.

Once again, we take the reviewer’s point and in the revised manuscript we have removed ‘limited’ and ‘efficient’ and simply described the concentrations of E2 and E3. Importantly, we show clearly in the revised manuscript that ‘low ubiquitylation conditions’ (2.5 nM E2 and 1 nM E3) lead to a single ubiquitin chain on CMG-Mcm7, whereas ‘high ubiquitylation conditions’ (25 nM E2 and 10 nM E3) correspond to 2-3 ubiquitin chains on CMG-Mcm7 (Figure 1—figure supplement 1). We show that p97-UFD1-NPL4 (without UBX proteins) disassembles the latter but not the former (Figure 2 and Figure 2—figure supplement 1). However, this does not require either multiple chains or linkages other than K48-linked ubiquitin chains (Figure 2—figure supplement 3). So the simplest interpretation is that extensively ubiquitylated CMG-Mcm7 favours productive binding to p97-UFD1-NPL4.

Given the inability of the p97 complex to unfold the CMG under "Low" ubiquitylation conditions, it might be worth asking whether increasing the amount of p97 complex in the reaction can overcome the barrier. This could potentially address issues related to reduced specific activity. It could also potentially address reduced Ub affinity for the p97 complex versus the Cdc48 complex.

We thank the reviewer for this suggestion – Figure 2—figure supplement 2 now shows that increasing the concentration of p97-UFD1-NPL, to a concentration 10X higher than needed to disassemble extensively ubiquitylated CMG, does not lead to efficient disassembly of CMG with a single chain of up to ~15 ubiquitins.

2. The data in Figure 3 are said to be based on CMG complexes that had up to ~12 ubiquitin conjugated to the Mcm7 subunit. This figure is also subject to the same issue as above – can the authors distinguish between one long chain versus some amount of multi-site ubiquitylation and shorter chains in this case?

The data now in Figure 1—figure supplement 1B+D show that the conditions used in Figure 3 involve predominantly one chain on CMG-Mcm7 (sometimes 2), almost exclusively K48-linked.

3. The deletion analysis in Figure 4 doesn't seem to example the extent to which the various mutants assemble with the p97 complex. One presumes a correlation, but this wasn't tested directly.

Figure 5—figure supplement 1 shows that the truncations of FAF1 and UBXN7 behave as expected (deletion of UBX blocks interaction with p97-UFD1-NPL4 but other truncated alleles can still bind the unfoldase).

One line 216, it is stated: "These findings indicated that the coiled-coil domain of FAF1 has a previously unanticipated role in stimulating the unfoldase activity of p97-UFD1-NPL4." However, it is possible that without this coiled-coil, the UBX domain doesn't associate effectively with the p97 complex. This would be sort of a trivial result, in that there isn't a "direct" role of the copied-coil, but it is simply required for the UBX domain to interact. Can the coiled-coil be replaced by an alternative coiled-coil sequence or is there something special about this coiled-coil?

Figure 5—figure supplement 1E-F shows that FAF1 UBX can interact with p97 without any requirement for the coiled coil, as predicted by past work (e.g. Kim et al., 2011, Proteins, 79, 2583-2587, which reported a crystal structure of the FAF1 UBX with the p97 N-terminus)

4. Although a bit hard to know for sure, it looks like the ubiquitylated forms of Mcm7 in the ES cell experiment (Figure 5) have much shorter chains than the experiments with yeast CMG – especially in 5b. Of course it is possible that this reflects the presence of DUB activity during the IP, which would tend to shorten the chains. But at face value, it looks like FAF1/UBXN7 can allow p97 to operate on very short chains (although one really cant distinguish this from multi-mono ubiquitylation).This raises questions about the overall conclusion. If p97 inhibitor increases the appearance of MCM7 conjugates (albeit with shorter chains) in the FAF1/UBXN7 knockout substantially, this suggest that p97 can disassemble shorter chains in cells without an adaptor OR that there is an additional UBX protein in cells that can do the job. Of course if the ubiquitylation levels are a result of DUB activity, that could explain it but it seems like this should be ruled out.

The disassembly of ubiquitylated CMG is blocked when cells are treated with p97 inhibitor, but it is likely that p97 inhibition also reduces the pool of free ubiquitin, leading indirectly to shorter ubiquitin chains on CMG-MCM7 (Figure 6B of the revised manuscript, discussed on lines 444-445). Similarly, previous studies with yeast (Maric et al., 2014), *C. elegans* (Sonneville et al., 2017) and mouse ES cells (Villa et al., 2021) have shown that p97 inhibition leads not only to accumulation of short ubiquitin chains on CMG-MCM7, but also to the accumulation of a significant fraction of CMG with non-ubiquitylated CMG-MCM7 (despite addition of DUB inhibitors such as propargylated ubiquitin to cell pellets and extracts), again most likely reflecting ubiquitin depletion upon inactivation of p97-UFD1-NPL4.

Reviewer #2 (Recommendations for the authors):1) The authors nail down the critical activity to the coiled-coil domain of the accessory factors FAF1/FAF2 (in addition to the UBX domain that binds p97). This is surprising and interesting because the UBA domains of FAF1 or FAF2 are not required. Frustratingly, it is not revealed what the function of the coiled-coil region is, which limits the mechanistic understanding. The authors speculate that the coiled coil region binds ubiquitin. This needs to be tested because it is a simple experiment. The comparison to Pup is inappropriate because Pup is not homologous to ubiquitin. Alternatively, the coiled coil region could modulate p97 or Ufd1-Npl4 binding activity. The question would be whether the coiled coil region binds p97 and/or Ufd1-Npl4. Technically, this can be tested.

Figure 5—figure supplement 1E-F shows that the coiled coil of FAF1 is not needed for interaction of the UBX with p97-UFD1-NPL4.

Figure 5—figure supplement 1C-D (FAF1-∆480) shows that the FAF1 coiled coil does not bind detectably to UFD1-NPL4 in the absence of p97.

We could not detect stable binding of the FAF1 coiled-coiled to K48-linked ubiquitin chains in vitro (Figure 5—figure supplement 3). However, the same is true for UFD1-NPL4-∆NZF (Figure 3—figure supplement 2), which nevertheless contains previously characterised low affinity ubiquitin-binding modules that we show are sufficient to support the disassembly of ubiquitylated CMG in reconstituted assays (Figure 3—figure supplement 2C, p97-UFD1-NPL4-∆NZF). Therefore, we cannot exclude that the FAF1 coiled coil also functions by binding to both p97-UFD1-NPL4 and ubiquitin, with ubiquitin binding being low affinity yet functionally relevant.

In our opinion, further mechanistic insight into the action of the coiled coil domain will require structural studies of early intermediates in the unfolding reaction, which are considerably beyond the scope of the present study.

2) The biochemical experiments show single time points. In contrast, the field has reached a point where kinetic data can be provided. That would allow to quantitatively define even smaller differences such as the deltaUBA or UIM effect in Figure 4C

Figure 2—figure supplement 1 presents a time course for the key experiment that illustrates the high ubiquitin threshold of human p97-UFD1-NPL4. Although we agree with the reviewer that further timecourses might have been applicable elsewhere in our study, the revision involves five months work leading to 11 new figures (plus new data in other figures). This meant that we had to prioritise our time and additional timecourse experiments were simply not possible.

3) The ES cell microscopy in Figure 5C needs to be quantified by an automated, unbiased pipeline.

We take the reviewer’s point but automated quantification is not easily applicable to such experiments, given the complex growth of mouse ES cells in three dimensions, which necessitates manual identification of cells to be counted. Moreover, we had to examine movie data manually to identify cells that entered mitosis and stayed in focus to allow quantification. The methods are now explained on lines 1061-1079.

4) Figure 6, cell viability and survival: The information gleaned from these experiments is questionable. Effects on colony formation are mostly mediated by MLN4924. The additional effect by the deletions is minor, and needs to be quantified and statistically validated, also to clarify whether it is additive or synthetic. The same applies to the viability assay. More importantly, it is impossible to connect the effects to replication termination. Firstly, there are many processes affected by the manipulations. Secondly, the MLN4924 treatment is so short that it will affect replication termination only in a small percentage of cells.

We have repeated the experiments formerly in Figure 6 (now presented in Figure 7) and the revised manuscript now presents quantification and statistical validation.

The reviewer questioned the connection of this experiment to DNA replication termination, but this experiment was not meant to be about DNA replication. p97-UFD1-NPL4 affect many areas of cell biology, and our data predict that UBX proteins such as UBXN7 and FAF1 (and others) will be important in many other processes (not just replication). The point of the experiment was to explore such broad significance of FAF1 and UBXN7, taking these two factors as representatives of UBX proteins that reduce the ubiquitin threshold of p97-UFD1-NPL4. Our data predicted that FAF1 and UBXN7 should become particularly important in diverse areas of cell biology, whenever ubiquitin chains on substrates are short. We tested this experimentally by impairing the activity of cullin ubiquitin ligases (as a major class of E3 enzymes in cells) by treatment with the neddylation inhibitor MLN4924. The data now in Figure 7B illustrate that cells lacking UBXN7 and FAF1 are more sensitive to MLN4924 than control cells, whilst Figure 7C shows that expression of FAF1 wild type (but not FAF1-∆UBX or FAF1-∆CoiledCoil) rescues this defect. Figure 7D shows that expression of UBXN7 rescues the MLN4924 sensitivity of UBXN7∆ FAF1∆ cells to the level of cells that just lack FAF1 (comparison with Figure 7B, FAF1∆).

[Editors' note: further revisions were suggested prior to acceptance, as described below.]

The manuscript has been improved but there are some remaining issues that need to be addressed, as outlined below:Reviewer #1 (Recommendations for the authors):Overall, the authors have done extensive work to try to address the previous reviewers' comments. The authors have addressed all of the specific comments, and in general, the results support the conclusions. One issue is that in Figure 6D, the p-value is only 0.09, which is not strongly significant. This may have to do with the assay itself. The majority of experiments employ ubiquitin smears to monitor p97 activity as it relates to the length dependence on the biological processes.

We now note in the legend to Figure 6D (lines 1592-1595) the weak statistical significance of the data and we note that this contrasts with Figure 6—figure supplement 4E, where the difference between UBXN7∆ FAF1∆ and cells lacking either UBXN7 or FAF1 is statistically significant.